# Constraining DALEC v2 using multiple data streams and ecological constraints: analysis and application.

Sylvain Delahaies[1], Ian Roulstone[1], and Nancy Nichols[2]

[1]Department of Mathematics, University of Surrey, Guildford, UK.
[2]Department of Mathematics, University of Reading, Reading, UK.

*Correspondence to:* S. Delahaies (s.b.delahaies@surrey.ac.uk)

**Abstract.** We use a variational method to assimilate multiple data streams into the terrestrial ecosystem carbon cycle model DALECv2. Ecological and dynamical constraints have recently been introduced to constrain unresolved components of this otherwise ill-posed problem. Here we recast these constraints as a multivariate Gaussian distribution to incorporate them into the variational framework and we demonstrate their benefit through a linear analysis. Using an adjoint method we study a linear approximation of the inverse problem: firstly we perform a sensitivity analysis of the different outputs under consideration, and secondly we use the concept of resolution matrices to diagnose the nature of the ill-posedness and evaluate regularisation strategies. We then study the non linear problem with an application to real data. Finally, we propose a modification to the model: introducing a spin-up period provides us with a built-in formulation of some ecological constraints which facilitates the variational approach.

## 1 Introduction

Carbon is a fundamental constituent of life and understanding its global cycle is a key challenge for the modelling of the Earth system. Through the processes of photosynthesis and respiration, ecosystems play a major role in the carbon cycle and thus in the dynamics of the global climate system. Our knowledge of the biogeochemical processes of ecosystems and an ever-growing amount of Earth observation systems can be combined using inverse modelling strategies to improve model predictions and uncertainty quantification.

The data assimilation linked ecosystem model (DALEC) is a simple box model for terrestrial ecosystems simulating a large range of processes occurring at different time scales from days to millennia. The work of Williams et al. (2005) established the benefit of using DALEC together with net ecosystem exchange of $CO_2$ (NEE) measurements in a Bayesian framework to estimate initial carbon stocks and model parameters, to improve flux predictions for ecosystem models, and to quantify

uncertainties. Inter-comparison experiments (Fox et al. (2009); Hill et al. (2012)) have then demonstrated the relative merit of various inverse modelling strategies using NEE and MODIS leaf area index observations: most results agreed on the fact that parameters and initial stocks directly related to fast processes were best estimated with narrow confidence intervals, whereas those related to slow processes were poorly estimated with very large uncertainties. Other studies have tried to overcome this difficulty by adding complementary data streams, see Richardson et al. (2010), or by considering longer observation windows, see Hill et al. (2012). Recently Bloom and Williams (2015) defined a set of ecological and dynamical constraints (EDCs) to reject unrealistic parameter combinations in the absence of additional data. However, to date very few systematic analysis has been carried out to explain the large differences among results.

As with many inverse problems, assimilating Earth observations into DALEC is an ill-posed problem: the model-observation operator which relates parameters and initial carbon stocks to the observations is rank deficient and not all variables can be estimated, or the model-observation operator is ill-conditioned and small observational noise may lead to a solution we can have little confidence in. Solving the problem amounts first to transforming it into a tractable problem in order to ensure a robust, meaningful and stable solution. This can be achieved by using regularisation techniques; the most popular one involves combining the observations and prior information, assuming it exists, through Bayesian inference. The choice of regularisation method depends on the nature of the problem and on the inverse modelling approach adopted.

So far, off-the-shelf methods such as ensemble Kalman filter (EnKF) and Monte Carlo Markov Chain (MCMC) were adopted to perform model-data fusion with DALEC. For its ability to accommodate non-linearity and any kind of probability distributions, the MCMC method, in the limit of a large number of samples, may be considered as the gold standard. However, despite being well suited for this type of small scale problem, the computational complexity of MCMC method makes it intractable for more complex situations. Here we adopt a variational approach (4DVAR) where a cost function measuring the mismatch between the model and observations is minimised using a gradient method based on the adjoint of the model. At Ameriflux sites (see http://ameriflux.lbl.gov/), we use MODIS monthly mean leaf area index (LAI) observations over a 13 year time window together with flux tower measurements of NEE and gross primary production (GPP). 4DVAR facilitates the diagnosis of the ill-posedness of the inverse problem: using model resolution matrices we can assess the resolution and stability properties of the observation operators and of the regularisation terms. We transcribe the EDCs into a novel variational framework and use some of this additional knowledge to estimate the otherwise undetermined variables. We consider a modification of the DALEC model by adding a spin-up period where carbon stocks are brought to equilibrium, this offers an alternative to including all the EDCs and helps reducing the confidence intervals for the predicted fluxes.

The paper is organised as follows. In section 2 we present DALECv2 and the observation streams used in this study; we review the EDCs introduced in Bloom and Williams (2015), and we perform a

sensitivity analysis of the different outputs of DALECv2 of interest for our experiments. In section 3 we recall basic principles of inverse theory from a Bayesian perspective, we introduce the variational formulation and we show how to incorporate the EDCs into this framework. Section 4 is devoted to a résumé of the linearised problem, using the tangent linear model, where the challenges of ill-posed problems and their regularisation can be explored in detail using simple linear algebra. Using a singular value decomposition we illustrate the effect of observational noise on ill-conditioned systems, and we investigate solution strategies from the point of view of resolution matrices. In section 5 we conduct a series of nonlinear inverse modelling experiments using multiple data streams and EDCs. In section 6 we modify DALECv2 to include a spin-up period which offers a built-in formulation of some EDCs, and then we reproduce the nonlinear experiments. In section 7 we discuss several extension to our manuscript and finally in section 8 we draw conclusions.

## 2 Model, constraints and observations

### 2.1 DALECv2

DALECv2 depicts a terrestrial ecosystem as a set of six carbon pools (labile $C_{\text{lab}}$, foliar $C_{\text{f}}$, wood $C_{\text{w}}$, root $C_{\text{r}}$, litterfall $C_{\text{l}}$ and soil organic matter $C_{\text{s}}$) linked via allocation fluxes. At a monthly time step the gross primary production (GPP) is calculated using the Aggregated canopy model (Williams et al. (1997)) as a nonlinear function of meteorological drivers (temperature, radiation , atmospheric $CO_2$ concentration), foliar carbon and foliar nitrogen. Following a mass conservation principle GPP is then allocated to the different carbon pools or released in the atmosphere via respiration. The schematic for DALECv2 is represented in Figure 1 and a complete description of the model can be found in Bloom and Williams (2015). DALECv2 combines the two previous DALEC-evergreen and DALEC-deciduous into a single model where the non-differentiable phenology process of DALEC-deciduous has been replaced with a differentiable process. DALECv2 is a nonlinear dynamical system and the carbon pools are dynamical variables parametrised by their initial values $C_0$ and by 17 parameters $\boldsymbol{p}$ whose range and description can be found in Table 1. The magnitudes and ranges of the parameters and the initial values vary drastically; therefore to avoid the computational problems caused by these different scales the variational methods will be formulated and implemented in terms of the log transformed variable $\boldsymbol{x} = \log([\boldsymbol{p}, \boldsymbol{C}_0])^T$. However in order to limit unnecessary notation and definition, in the remainder of this paper $\boldsymbol{p}$ and $\boldsymbol{C}_0$ will stand for their $\log$ transform.

The meteorological drivers are extracted from $0.125°\text{x}0.125°$ERA-interim reanalysis datasets. For the purpose of our inverse modelling experiments we use four different observation streams: LAI, NEE, GPP and RESP. LAI monthly mean observations for Ameriflux sites are extracted from MOD15A2 LAI 8-day version 005 1km-resolution product. These observations together with the meteorological drivers are provided by A. Bloom and J. Exbrayat: details about their construction can be found in Bloom and Williams (2015). At Ameriflux sites we use the level 4 data product

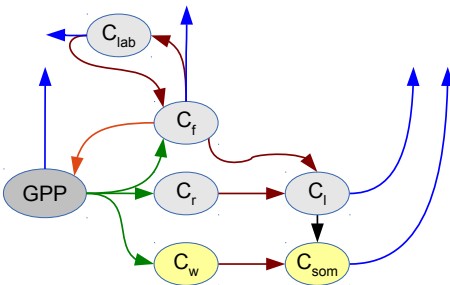

**Figure 1.** DALECv2 links the carbon pools (C) via allocation fluxes (green), litterfall fluxes (red), decomposition (black). Respiration is represented by the blue arrows. The orange arrow represents the feedback of foliar carbon to gross primary production (GPP).

**Table 1.** DALECv2 dynamical variables and parameters with their respective range. The units of the non dimensionless quantities are given in brackets.

| Label | Variable | description | range |
|---|---|---|---|
| $\boldsymbol{C}_0(1)$ | $C_{\text{lab}}$ | initial labile C pool ($\text{gCm}^{-2}$) | 20 - 2000 |
| $\boldsymbol{C}_0(2)$ | $C_{\text{f}}$ | initial foliar C pool ($\text{gCm}^{-2}$) | 20 - 2000 |
| $\boldsymbol{C}_0(3)$ | $C_{\text{r}}$ | initial fine root C pool ($\text{gCm}^{-2}$) | 20 - 2000 |
| $\boldsymbol{C}_0(4)$ | $C_{\text{w}}$ | initial above and below ground woody C pool ($\text{gCm}^{-2}$) | $100 - 10^5$ |
| $\boldsymbol{C}_0(5)$ | $C_{\text{l}}$ | initial litter C pool ($\text{gCm}^{-2}$) | 20 - 2000 |
| $\boldsymbol{C}_0(6)$ | $C_{\text{s}}$ | initial soil organic matter C pool ($\text{gCm}^{-2}$) | $100 - 2 \times 10^5$ |
| $p_1$ | $\theta_{\text{min}}$ | Litter mineralisation rate ($\text{day}^{-1}$) | $10^{-5} - 10^{-2}$ |
| $p_2$ | $f_{\text{a}}$ | autotrophic respiration fraction | 0.3 - 0.7 |
| $p_3$ | $f_{\text{f}}$ | fraction of GPP allocated to $C_{\text{f}}$ | 0.01 - 0.5 |
| $p_4$ | $f_{\text{r}}$ | fraction of GPP allocated to $C_{\text{r}}$ | 0.01 - 0.5 |
| $p_5$ | $c_{\text{lf}}$ | Annual Leaf Loss Fraction (season) | 1 - 8 |
| $p_6$ | $\theta_{\text{w}}$ | $C_{\text{w}}$ turnover rate ($\text{day}^{-1}$) | $2.5 \times 10^{-5} - 10^{-3}$ |
| $p_7$ | $\theta_{\text{r}}$ | $C_{\text{r}}$ turnover rate ($\text{day}^{-1}$) | $10^{-4} - 10^{-2}$ |
| $p_8$ | $\theta_{\text{l}}$ | $C_{\text{l}}$ turnover rate ($\text{day}^{-1}$) | $10^{-4} - 10^{-2}$ |
| $p_9$ | $\theta_{\text{s}}$ | $C_{\text{s}}$ turnover rate ($\text{day}^{-1}$) | $10^{-7} - 10^{-3}$ |
| $p_{10}$ | $\Theta$ | temperature dependence exponent factor | 0.018 - 0.08 |
| $p_{11}$ | $c_{\text{eff}}$ | Canopy Efficiency Parameter | 10 - 100 |
| $p_{12}$ | $d_{\text{onset}}$ | Leaf Onset Day (day) | 1 - 365 |
| $p_{13}$ | $f_{\text{l}}$ | fraction of GPP allocated to $C_{\text{lab}}$ | 0.01 - 0.5 |
| $p_{14}$ | $c_{\text{ronset}}$ | $C_{\text{lab}}$ release period (day) | 10 - 100 |
| $p_{15}$ | $d_{\text{fall}}$ | Leaf Fall Day (day) | 1 - 365 |
| $p_{16}$ | $c_{\text{rfall}}$ | Leaffall period (day) | 20 - 150 |
| $p_{17}$ | $c_{\text{lma}}$ | Leaf Mass per area ($\text{gCm}^{-2}$) | 10 - 400 |

(available at http://cdiac.ornl.gov/ftp/ameriflux/data/Level4/), which provides monthly means for NEE and GPP. NEE and GPP are then used to define total respiration (RESP) as RESP=NEE+GPP. The meteorological drivers span over a period of twelve years from 2001 to 2013. LAI observations are available during the full period but for NEE and GPP, and thus RESP, shorter records are available depending on the Ameriflux site. In this study we consider the Morgan Monroe state forest located in Indiana, US (39.3,-86.4). This Ameriflux site is composed in majority of mixed hardwood broadleaf deciduous trees and classifies as a humid subtropical climate.

In the remainder of the paper the main focus is on the vector $\boldsymbol{x} = \log([\boldsymbol{p}, \boldsymbol{C}_0])^T$: in section 2.3 first where we investigate the sensitivity of different outputs with respect to $\boldsymbol{x}$ and its components, and then in subsequent sections where $\boldsymbol{x}$ is estimated using inverse methods. The vector $\boldsymbol{x}$, denoting fixed quantities as initial conditions and parameters for the dynamical system DALECv2, is seen as the variable from the point of view of sensitivity analysis and inverse modelling and therefore its components will be referred to as state variables, input variables or parameters interchangeably throughout the manuscript.

## 2.2 Ecological constraints

Over the last decade many inverse modelling studies have used NEE measurements from the fluxnet network, together with other types of observations when available, to provide information about processes controlled by parameters with respect to which NEE is weakly sensitive. Though it contains an ever-increasing amount of information, the flux tower network only provides sparse coverage of terrestrial ecosystems. On the other hand, despite a good spatial and temporal coverage, MODIS LAI monthly mean observations only constrain a limited set of DALECv2 state variables, and additional information is required in order to regularise the ill-posed problem and obtain a meaningful solution. Additional information can be obtained by imposing priors on the variables or by adding other observation streams (biomass, soil organic matter, ...). As an alternative, Bloom and Williams introduced a set of constraints, referred to as ecological and dynamical constraints (EDCs). These constraints, detailed in Bloom and Williams (2015) can be divided into two groups: static and dynamic constraints. The static constraints which directly impose conditions on the parameters are:

- turnover rates constraints which ensure that turnover rates ratios are consistent with knowledge of the carbon pools residence times.

$$\text{EDC}_1: \quad p_9 < p_8, \tag{1}$$

$$\text{EDC}_2: \quad p_9 < p_1, \tag{2}$$

$$\text{EDC}_3: \quad p_6 < 1/(p_5 \times 365.25), \tag{3}$$

$$\text{EDC}_4: \quad p_7 > p_9 \exp p_{10} \bar{T}, \tag{4}$$

$$\text{EDC}_5: \quad p_{12} + 45 < p_{15}, \tag{5}$$

where $\bar{T}$ denotes the mean temperature within the drivers time window. $EDC_4$ is a modification to the constraint proposed in Bloom and Williams (2015), it is currently used in the CARDAMON framework (http://www.geos.ed.ac.uk/homes/mwilliam/CARDAMOM.html).

– Root-foliar allocation which allows for a strong correlation between parameters controlling allocation to foliage and roots.

$$EDC_6: \quad f_{root} < 5(f_{fol} + f_{lab}), \tag{6}$$

$$EDC_7: \quad f_{fol} + f_{lab} < 5f_{root}, \tag{7}$$

where the allocation fractions $f_{fol}$, $f_{lab}$ and $f_{root}$ are defined by

$$f_{auto} = p_2, \tag{8}$$

$$f_{fol} = (1 - f_{auto})p_3, \tag{9}$$

$$f_{lab} = (1 - f_{auto} - f_{fol})p_{13}, \tag{10}$$

$$f_{root} = (1 - f_{auto} - f_{fol} - f_{lab})p_4. \tag{11}$$

The dynamic constraints, for which a model run is performed to define attractors, limit the application of the model to ecosystems with no major recent disturbance. They are defined by:

– Root-foliar mean dynamics

$$EDC_8 : \bar{C}_r < 5\bar{C}_f, \tag{12}$$

$$EDC_9 : \bar{C}_f < 5\bar{C}_r, \tag{13}$$

where $\bar{C}_f$ and $\bar{C}_r$ denote the mean of $C_f$ and $C_r$ over the simulation period.

– Yearly carbon pools growth rate is limited to 10%.

$$EDC_{10-15}: \quad \bar{C}^n/\bar{C}^1 < 1 + \zeta(n-1)/10, \tag{14}$$

where for each pool $\bar{C}^i$ denotes the mean carbon pool size over year $i$ and the growth factor $\zeta$ is set to 0.1.

– Carbon pools are not expected to show rapid exponential decay; therefore parameter sets are required to satisfy the condition that the half-life period of carbon pools is more than three years.

$$EDC_{16-21}: \quad \gamma < 3 \times 365/\log 2. \tag{15}$$

The trajectory of each carbon pool is approximated using an exponential decay curve $a + b\exp\gamma t$ where $a$, $b$ and $\gamma$ are the fitted exponential decay parameters and t the time variable, in days in this case.

– Carbon pools are expected to be within an order of magnitude of a steady state attractor.

$$\text{EDC}_{22-29}: \quad C_0/10 < C^\infty < 10C_0, \tag{16}$$

where for each of the carbon pools $C_\text{s}$, $C_\text{l}$, $C_\text{w}$ and $C_\text{r}$, $C_0$ denotes the initial state and $C^\infty$ denotes the steady state attractor defined by

$$C_\text{som}^\infty = \frac{(f_\text{wood} + (f_\text{fol} + f_\text{root} + f_\text{lab})p_1)\bar{G}}{(p_1 + p_9)p_8 \exp \bar{T}p_{10}}, \tag{17}$$

$$C_\text{lit}^\infty = \frac{(f_\text{fol} + f_\text{root} + f_\text{lab})\bar{G}}{p_9 \exp \bar{T}p_{10}}, \tag{18}$$

$$C_\text{wood}^\infty = \frac{f_\text{wood}\bar{G}}{p_6}, \tag{19}$$

$$C_\text{root}^\infty = \frac{f_\text{root}\bar{G}}{p_7}, \tag{20}$$

where $\bar{G}$ denotes the mean gross primary production and $f_\text{wood}$, $f_\text{som}$ and $f_\text{lit}$ are given by

$$f_\text{wood} = 1 - f_\text{auto} - f_\text{fol} - f_\text{lab} - f_\text{root}, \tag{21}$$

$$f_\text{som} = f_\text{wood} + (f_\text{root} + f_\text{lab} + f_\text{fol})p_1/(p_1 + p_8), \tag{22}$$

$$f_\text{lit} = (f_\text{root} + f_\text{lab} + f_\text{fol}). \tag{23}$$

To the original EDCs, we found useful to add the three following constraints:

$$\text{EDC}_{30}: \quad \text{LAI(summer)} < \alpha, \; \alpha > 0, \tag{24}$$

$$\text{EDC}_{31}: \quad \text{LAI(final day)} > 0, \tag{25}$$

$$\text{EDC}_{32,33}: \quad -\beta < \text{E[NEE]} < \beta, \; \beta > 0, \tag{26}$$

where $\alpha$ and $\beta$ are real constants that need to be adjusted, LAI(summer) denotes the modelled LAI during summer and LAI(final day) denotes the modelled LAI at the end of the model run. These new constraints guarantee that LAI and the mean NEE remain between realistic bounds.

Bloom and Williams demonstrated the efficiency of incorporating EDCs using a Monte Carlo method to improve parameter estimates and NEE predictions. We propose an approach to apply these extra

constraints within a variational framework.

### 2.3 Sensitivity analysis

Sensitivity analysis studies how the variations of the output $\boldsymbol{h}$ of a model can be attributed to variations of the input variables $x_i$. Such information is crucial for model design, inverse modelling and reduction of complex nonlinear models. A global sensitivity analysis for DALEC was recently per-

195 formed in Safta et al. (2015), here we consider a local approach where first order derivatives are used to build sensitivity indices that help understanding the influence of input variables on the output.

We denote by $h_t$ the function that maps $\boldsymbol{x} = \log(\boldsymbol{p}, \boldsymbol{C}^0)$ to the value of an output of the model

(here LAI, NEE, GPP and RESP) at time $t$ and we denote by $\boldsymbol{h} = (h_{t_1}, ..., h_{t_N})$ the time series of the model output. Following Zhu and Zhuang (2014), we consider the mean normalised sensitivity (MNS) defined by

$$s_i = \mathrm{E}\left( \frac{\partial \boldsymbol{h}}{\partial x_i} \left| \frac{\sigma_i}{\sigma_h} \right| \bigg/ \sum_j \left| \frac{\partial \boldsymbol{h}}{\partial x_j} \frac{\sigma_j}{\sigma_h} \right| \right), \tag{27}$$

where $\mathrm{E}(\cdot)$ denotes the average of the time series. The scalars $\sigma_i$ and $\sigma_h$ denote the parameter variance, set as 40% of the parameter range, and the variance of the output respectively. The partial derivatives are computed using the adjoint derived using the method described in Giering and Kaminski (1998). The MNS $s_i$ is a dimensionless number that allows us to compare between parameters. We consider the Morgan Monroe State Forest over a thirteen year period. We sample 100 parameter sets satisfying the ecological constraints. For each parameter set we compute the MNS for DALEC simulated mean fluxes LAI, NEE. In Figure 2 parameters are ranked with respect to their mean MNS. We see that for LAI only 12 out of the 23 variables are sensitive, namely $p_5$, $p_{17}$, $p_2$, $p_{13}$, $p_{11}$, $p_{15}$, $p_{16}$, $C_{\mathrm{f}}$, $p_{12}$, $p_3$, $C_{\mathrm{lab}}$ and $p_{14}$. Therefore using LAI only in an inverse modelling experiment provides, at best, information about those twelve sensitive variables. For NEE we see that all variables are sensitive. Sensitivity analysis for GPP shows similar characteristics with LAI and so does RESP with NEE. For the four outputs under consideration (LAI, NEE, GPP and RESP) the most sensitive variables are the autotrophic respiration, $p_2$, the annual leaf loss fraction, $p_5$, the leaf mass per area, $p_{17}$, the fraction of GPP allocated to labile pool, $p_{13}$, the nitrogen use efficiency, $p_{11}$, and the leaf fall day $p_{15}$.

Here our focus is on the mean of the time series of DALEC fluxes (LAI, NEE) over a thirteen year period. Finer analysis could be carried out by looking at seasonal aspects of the carbon cycle, identifying what variables are the most sensitive at certain time of the year for example as studied in Safta et al. (2015).

## 3 Data assimilation

In this section we introduce concepts and methods that allow for a close mathematical scrutiny of inverse problems and we present the variational method that will be used for applications.

### 3.1 Ill posed problem

A generic inverse problem consists in finding a $n$-dimensional state vector $\boldsymbol{x}$ such that

$$\boldsymbol{h}(\boldsymbol{x}) = \boldsymbol{y}, \tag{28}$$

for a given $N$-dimensional observation vector $\boldsymbol{y}$, including random noise, and a given model $\boldsymbol{h}$. In the remainder of the paper the terms state vector, state variable, input variable or parameters will be used interchangeably to denote the vector $\boldsymbol{x}$ to be estimated using inverse methods and defined

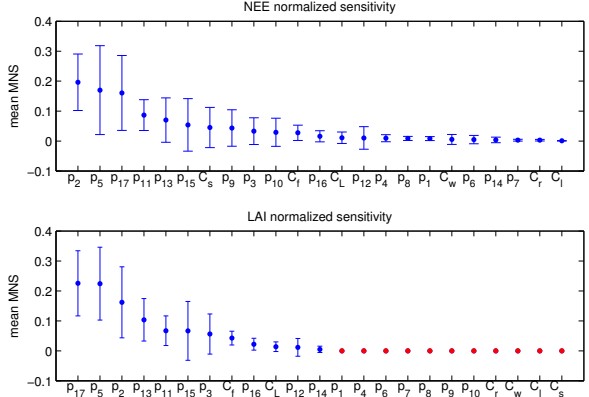

**Figure 2.** Mean normalised sensitivities : 100 parameter sets satisfying EDCs are sampled at the Morgan Monroe State Forest. Parameters are ranked in decreasing order according to their sensitivity, the blue dots represent the mean of the MNS (dimensionless quantity) and the intervals represent 1-sigma error bars; the red dots correspond to null sensitivity.

in the previous section as $\boldsymbol{x} = \log([\boldsymbol{p}, \boldsymbol{C}])^T$. The problem is well posed in the sense of Hadamard (1923) if the three following conditions hold: 1) there exists a solution, 2) the solution is unique and 3) the solution depends continuously on the input data. If at least one of these conditions is violated the problem is said to be ill-posed. The inverse problem (28) is often ill-posed, and a regularisation method is required to replace the original problem with a well-posed problem. Solving (28) amounts to 1) constructing a solution $\boldsymbol{x}$, 2) assessing the validity of the solution, 3) characterising its uncertainty. Each inverse problem has its own features which need to be understood in order to characterise properly the solution and its uncertainty.

### 3.2 Bayesian inference: 4DVAR

Inverse problems are generally presented in a probabilistic framework where most methods can be expressed through a Bayesian formulation. The Bayesian approach provides a full characterisation of all possible solutions, their relative probabilities and uncertainties.

From Bayes' theorem the probability density function (pdf) of the model state $\boldsymbol{x}$ given the set of observations $\boldsymbol{y}$, $p(\boldsymbol{x}|\boldsymbol{y})$, is given by

$$p(\boldsymbol{x}|\boldsymbol{y}) \propto p(\boldsymbol{y}|\boldsymbol{x})p(\boldsymbol{x}), \tag{29}$$

where $p(\boldsymbol{y}|\boldsymbol{x})$ is the pdf of the observations given $\boldsymbol{x}$ and $p(\boldsymbol{x})$ is the prior pdf of $\boldsymbol{x}$. A special case is given when $p(\boldsymbol{y}|\boldsymbol{x})$ and $p(\boldsymbol{x})$ are Gaussian pdf given by

$$p(\boldsymbol{x}) = \exp\left[-\frac{1}{2}(\boldsymbol{x} - \boldsymbol{x}_0)^T \mathbf{B}^{-1}(\boldsymbol{x} - \boldsymbol{x}_0)\right], \tag{30}$$

and

$$p(\boldsymbol{y}|\boldsymbol{x}) = \exp\left[-\frac{1}{2}(\boldsymbol{h}(\boldsymbol{x}) - \boldsymbol{y})^T \mathbf{R}^{-1}(\boldsymbol{h}(\boldsymbol{x}) - \boldsymbol{y})\right], \tag{31}$$

where $\mathbf{B}$ is the covariance matrix of the prior term $\boldsymbol{x}_0$, and $\mathbf{R}$ is the covariance matrix of the observation error. When the operator $\boldsymbol{h}$ is linear then the posterior pdf $p(\boldsymbol{x}|\boldsymbol{y})$ is Gaussian and thus fully characterised by its mean and covariance matrix. The mean is obtained by minimising the modulus of the log of the joint probability distribution, that is the cost function $J$ given by

$$J(\boldsymbol{x}) = J_0(\boldsymbol{x}) + J_y(\boldsymbol{x}) = \frac{1}{2}\|\boldsymbol{x} - \boldsymbol{x}_0\|_\mathbf{B}^2 + \frac{1}{2}\|\boldsymbol{h}(\boldsymbol{x}) - \boldsymbol{y}\|_\mathbf{R}^2. \tag{32}$$

Many methods can be considered to minimise this cost function. A Monte Carlo method is employed in Bloom and Williams (2015). Here we use a variational approach which applies a gradient based method where the gradient is given by

$$\boldsymbol{\nabla} J = \mathbf{B}^{-1}(\boldsymbol{x} - \boldsymbol{x}_0) + \mathbf{H}^T \mathbf{R}^{-1}(\boldsymbol{h}(\boldsymbol{x}) - \boldsymbol{y}), \tag{33}$$

with $\mathbf{H}^T$ denoting the adjoint operator. The covariance matrix of the solution, $\mathbf{C}$, is given by the inverse of the hessian of the cost function

$$\mathbf{C} = [\text{Hess}(J)]^{-1} = \left[\mathbf{B}^{-1} + \mathbf{H}^T \mathbf{R}^{-1} \mathbf{H}\right]^{-1}. \tag{34}$$

When the observation operator $\boldsymbol{h}$ is non-linear, the cost function $J$ can have multiple local minima and the posterior pdf may no longer be a Gaussian pdf. However, locally, the pdf $N(\tilde{\boldsymbol{x}}, \mathbf{C})$, where $\mathbf{C}$ is given by equation (34) evaluated at a minimum $\tilde{\boldsymbol{x}}$, provides a Gaussian approximation of the posterior pdf $p(\boldsymbol{x}|\boldsymbol{y})$.

The first term in the cost function (32) is a regularisation term encoding the Gaussian prior $p(\boldsymbol{x})$. As we will show in the next sections the problem of assimilating EO observations (LAI,GPP,NEE,RESP) into DALEC is a highly ill-posed problem and regularisation is required. The sensitivity analysis of section 2.3 showed that LAI and GPP are not sensitive to all variables. Moreover all observations streams show very small sensitivities to some variables. Therefore, as will be illustrated in section 4.1, the solution (if any) is likely to be subject to large uncertainties. Apart from a couple of extensively studied sites, our prior knowledge about the variables is so far limited to their upper and lower bounds given in Table 1. As performed in Zhu and Zhuang (2014), it is a common practice to use this information to define a Gaussian prior $p(\boldsymbol{x}) \sim N(\boldsymbol{x}_0, \mathbf{B})$, where $\boldsymbol{x}_0$ is given by the centre of the variables ranges and $\mathbf{B}$ is the diagonal matrix whose diagonal elements are the squares of 40% of the variables ranges. While using this kind of regularisation is necessary to ensure any solution at all when no better source of information is available, this introduces some biases in the solution. The EDCs introduced by Bloom and Williams (2015) provide new prior information about the variables. One of the purposes of this paper is to incorporate the EDCs as a regularisation term within 4DVAR. In the next section we propose a strategy to achieve this goal.

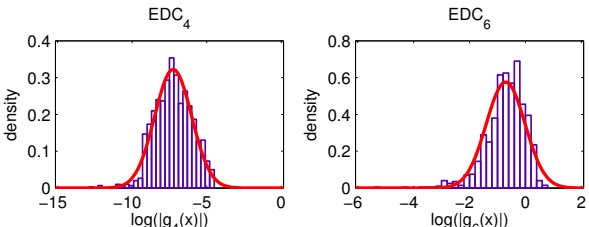

**Figure 3.** distribution and Gaussian fit for $EDC_4$ and $EDC_6$.

### 3.3 EDCs and 4DVAR

Incorporating the EDCs from an optimisation point of view can be easily performed by considering an inequality constraint optimisation problem where we aim at solving

$$\min_{\boldsymbol{x}} J_y(\boldsymbol{x}) \quad \text{subject to} \quad \boldsymbol{l} < \boldsymbol{x} < \boldsymbol{u} \text{ and } \boldsymbol{g}(\boldsymbol{x}) < \boldsymbol{0},$$

where $\boldsymbol{g}$ is the nonlinear operator defining the EDCs described in Section 2.2, and $\boldsymbol{l}$ and $\boldsymbol{u}$ are the lower and upper bounds defined in Table 1. This approach provides an efficient, robust and quick strategy to find an acceptable solution; however, stability properties are not easily determined, see Roese-Koerner et al. (2012).

We are seeking for a multivariate Gaussian distribution that would encode the EDCs. At a forest site, we start by sampling the parameter space to obtain an ensemble of 1000 parameter sets satisfying the EDCs; each parameter set $\boldsymbol{x}$ is randomly created and required to satisfy $\boldsymbol{g}(\boldsymbol{x}) < \boldsymbol{0}$. We denote this ensemble by $\mathcal{X}_{\text{EDCs}}$. For most parameters, the sampling gives rise to undetermined pdfs which can certainly not be represented by Gaussian pdfs. However inspecting the distribution $\boldsymbol{g}(\boldsymbol{x})$, for all $\boldsymbol{x}$ in

$\mathcal{X}_{\text{EDCs}}$, we see that the distribution $\log(\boldsymbol{g}(\boldsymbol{x}))$ can be fairly accurately approximated by a multivariate Gaussian pdfs $N(\boldsymbol{c}, \boldsymbol{\Sigma})$ where $\boldsymbol{c}$ denotes the mean of the distribution $\log(\boldsymbol{g}(\boldsymbol{x}))$ and $\boldsymbol{\Sigma}$ denotes its covariance matrix. As an example Figure 3 shows the marginals $\log(g_4(\boldsymbol{x}))$ and $\log(g_6(\boldsymbol{x}))$, corresponding to the EDCs 4 and 6 respectively, together with a Gaussian fit.

Using Bayes' Theorem we can then write

$$p(\boldsymbol{x}|\boldsymbol{y}, \boldsymbol{c}) \propto p(\boldsymbol{y}|\boldsymbol{x})p(\boldsymbol{c}|\boldsymbol{x})p(\boldsymbol{x}). \tag{35}$$

Finding a Gaussian approximation for $p(\boldsymbol{x}|\boldsymbol{y}, \boldsymbol{c})$ amounts then to minimising the cost function

$$J(\boldsymbol{x}) = \frac{1}{2}\|\boldsymbol{h}(\boldsymbol{x}) - \boldsymbol{y}\|_{\mathbf{R}}^2 + \frac{1}{2}\|\log(\boldsymbol{g}(\boldsymbol{x})) - \boldsymbol{c}\|_{\boldsymbol{\Sigma}}^2 + \frac{1}{2}\|\boldsymbol{x} - \boldsymbol{x}_0\|_{\mathbf{B}}^2, \tag{36}$$

The gradient of $J$ is given by

$$\boldsymbol{\nabla} J(\boldsymbol{x}) = \mathbf{H}^T \mathbf{R}^{-1}(\boldsymbol{h}(\boldsymbol{x}) - \boldsymbol{y})$$

$$+ \frac{1}{\boldsymbol{g}(\boldsymbol{x})}\mathbf{G}^T \boldsymbol{\Sigma}^{-1}(\log(\boldsymbol{g}(\boldsymbol{x})) - \boldsymbol{c}) + \mathbf{B}^{-1}(\boldsymbol{x} - \boldsymbol{x}_0),$$

and the hessian of the cost function can be approximated by

$$\mathcal{H} = \mathbf{H}^T \mathbf{R}^{-1} \mathbf{H} + \frac{1}{(\boldsymbol{g}(\boldsymbol{x}))^2} \mathbf{G}^T \boldsymbol{\Sigma}^{-1} \mathbf{G} + \mathbf{B}^{-1}, \tag{37}$$

evaluated at the minimiser $\tilde{\boldsymbol{x}}$. The operator $\mathbf{G}^T$ denotes the adjoint of the tangent linear model $\mathbf{G}$ whose key ingredient is given by the adjoint of DALECv2. The approximation of $p(\boldsymbol{x}|\boldsymbol{y}, \boldsymbol{c})$ is then given by the Gaussian distribution $N(\tilde{\boldsymbol{x}}, \mathcal{H}^{-1})$. In Section 5 we will perform experiments using real data to validate this approach.

## 4   Linear analysis

Considerable theoretical insights into the nature of the inverse problem, and the ill-posedness, can be obtained by studying a linearisation of the operator $\boldsymbol{h}$. A first approximation to the inverse problem consists in finding a perturbation $\boldsymbol{z}$ which best satisfies the linear equation

$$\mathbf{H}\boldsymbol{z} = \boldsymbol{d}, \tag{38}$$

where $\mathbf{H}$ is the tangent linear operator for $\boldsymbol{h}$ and $\boldsymbol{d}$ is a perturbation of the observations. The linear operator $\mathbf{H}$ is commonly referred to as the *observability matrix* (see Johnson et al. (2005)). The least squares formulation of this problem is to solve the optimisation problem

$$\min_{\boldsymbol{z}} J(\boldsymbol{z}) = \min_{\boldsymbol{z}} \frac{1}{2} \|\mathbf{H}\boldsymbol{z} - \boldsymbol{d}\|^2. \tag{39}$$

The minimisation can be performed using an iterative method such as the conjugate gradient method, where the gradient is given by

$$\boldsymbol{\nabla} J = \mathbf{H}^T (\mathbf{H}\boldsymbol{z} - \boldsymbol{d}). \tag{40}$$

The inverse hessian of the cost function, $(\mathbf{H}^T\mathbf{H})^{-1}$, gives the covariance matrix of the least squares solution. In the next section we consider a direct solution method based on the singular value decomposition of the operator $\mathbf{H}$, which allows us to investigate the nature of the ill-posedness of the problem. We illustrate regularisation using a truncated singular value decomposition.

### 4.1   Singular value decomposition

We consider a singular value decomposition of $\mathbf{H}$ of the form

$$\mathbf{H} = \mathbf{U}\mathbf{S}\mathbf{V}^T, \tag{41}$$

where $\mathbf{U}$ is a $N \times N$ unitary matrix, $\mathbf{V}$ is a $n \times n$ unitary matrix and $\mathbf{S}$ is the $N \times n$ diagonal matrix whose diagonal elements are the singular values $s_1 \geq \cdots \geq s_n \geq 0$. Using this decomposition, the solution $\boldsymbol{z}_{\mathrm{LS}}$ to (39) can be written as

$$\boldsymbol{z}_{\mathrm{LS}} = \mathbf{V}\mathbf{S}^\dagger \mathbf{U}^T \boldsymbol{y} = \mathbf{H}^\dagger \boldsymbol{y}. \tag{42}$$

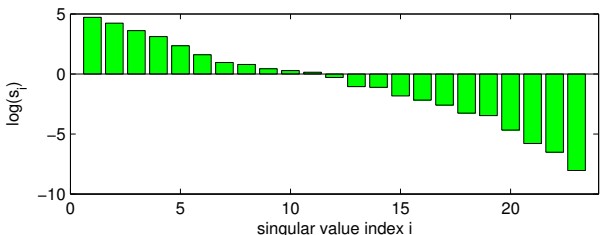

**Figure 4.** Singular values of the observability matrix for NEE (log scale)

The matrix $\mathbf{H}^\dagger = \mathbf{V}\mathbf{S}^\dagger\mathbf{U}^T$ is the *pseudo-inverse* of $\mathbf{H}$ where $\mathbf{S}^\dagger$ is the diagonal matrix obtained by transposing $\mathbf{S}$ and replacing the non zero elements by their inverse $s_i^{-1}$. The covariance of the solution is given by

$$\text{Cov}(\boldsymbol{z}_{\text{LS}}) = \mathbf{H}^{\dagger T}\mathbf{H}^\dagger. \tag{43}$$

Much can be learned about the stability of the solution (42) by inspecting the singular values of $\mathbf{H}$. Assuming that $\mathbf{H}$ is full rank, it can be shown, (see Golub and Van Loan (1996)), that the relative error in the solution, defined as the left hand side of the above inequality, is bounded by

$$\frac{\|\boldsymbol{z}_{\text{LS}} - \boldsymbol{z}_0\|}{\|\boldsymbol{z}_0\|} \leq \kappa(\mathbf{H})\frac{\|\boldsymbol{\epsilon}\|}{\|\boldsymbol{d}\|}, \tag{44}$$

where $\kappa(\mathbf{H})$ is the condition number of $\mathbf{H}$ defined by $\kappa(\mathbf{H}) = s_1/s_n$, $\boldsymbol{z}_0$ denotes the truth (possibly unknown) and $\boldsymbol{\epsilon}$ represents observational noise. When the condition number is large the matrix is said to be ill-conditioned, the problem is ill-posed and the solution (42) is unstable: small perturbations to the system can lead to very large perturbations in the solution.

## 4.2 Stability for NEE operator

As an example we consider the problem of assimilating NEE observations into DALECv2 to estimate model parameters and initial conditions at Morgan Monroe State forest. We linearise equation (28) about a point $\boldsymbol{x}^*$ satisfying the EDCs, we form the observability matrix $\mathbf{H}$ and compute its singular value decomposition. The singular values, shown in figure 4, reveal a condition number of order $10^5$. For a signal-to-noise ratio, namely $\|\boldsymbol{\epsilon}\|/\|\boldsymbol{d}\|$, of magnitude $0.1$, inequality (44) gives an upper bound for the relative error in the solution of order $10^4$, which does not give much credit to the least square solution. How sharp is this bound? Are we overestimating the error? To answer these questions we create a set of noisy observations with noise variance $\sigma = 0.1$ and we compute the solution (42). The relative error for each component of the solution, $\eta_i$, and the variance $\nu_i$, are given in Table 2. Despite a relatively good match between the modelled NEE perturbations and the observations, as shown in figure 5, the results of Table 2 show very large relative errors and variances for most variables. Moreover these results are in agreement with the results of REFLEX: parameters directly

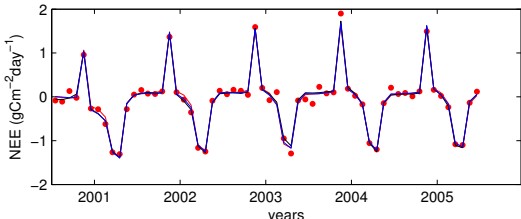

**Figure 5.** Solution of the linearised inverse problem for NEE. The red points represent the observations, the red curve is the true trajectory, the green curve is the trajectory obtained using the unstable solution and the blue curve is obtained using the TSVD solution.

linked to foliage and GPP are better estimated than parameters related to allocation to and turnover of fine root/wood. The results of Table 2 reflect the sensitivity analysis shown on figure 2. The variables with respect to which NEE is the most (resp. least) sensitive are the less (resp. more) affected by the noise.

To reduce the impact of observational noise on the solution, regularisation is required. The *truncated singular value decomposition* (TSVD) is a simple and popular method for regularisation. TSVD consists in truncating the pseudo-inverse in equation (42) in order to remove the smallest singular values, the most affected by the noise. The solution $\boldsymbol{z}^{(k)}$ is then given by

$$\boldsymbol{z}^{(k)} = \mathbf{V}_k \mathbf{S}_k^\dagger \mathbf{U}_k^T \boldsymbol{y} = \mathbf{H}_k^\dagger \boldsymbol{y}, \tag{45}$$

where $k$ is the truncation rank, $\mathbf{S}_k$, $\mathbf{U}_k$ and $\mathbf{V}_k$ are the rectangular matrices formed by the first $k$ columns of $\mathbf{S}$, $\mathbf{U}$ and $\mathbf{V}$. The covariance of the solution is given by

$$\mathrm{Cov}(\boldsymbol{z}^{(k)}) = \mathbf{V}_k (\mathbf{S}_k^\dagger)^{-2} \mathbf{V}_k^T. \tag{46}$$

The truncation rank $k$ can be chosen using the L-curve method. The L-curve is a log-log plot of the norm of the solution $\|\boldsymbol{z}^{(k)}\|$ against the norm of the residual $\|\mathbf{H}\boldsymbol{z}^{(k)} - \boldsymbol{d}\|$ parametrised by the regularisation parameter $k$. The optimal parameter corresponds to the point of maximum curvature of the L-curve. Further details on the L-curve method can be found in Hansen and O'Leary (1993). In our example with NEE we use Hansen's regularisation tools, see Hansen (2007), to perform the TSVD method. The truncation rank obtained using the L-curve method is $k = 7$. The last three columns of Table 2, presenting the TSVD solution, the relative error of each components and the variances, can be compared with the results for the unstable solution. Whereas the relative errors in the unstable solution range from $5.3 \times 10^{-2}$ to $3.8 \times 10^4$ the relative errors in the regularised solution range from $5.3 \times 10^{-2}$ to 5.1. We see that TSVD has the effect of keeping small the variables that cannot be estimated correctly. As previously stated the results of the regularisation can be related to the sensitivity analysis depicted on Figure 2: TSVD prevents the variables with respect to which NEE is the least sensitive from growing unbounded.

**Table 2.** Results of the linear inverse problem showing: 1- the solution components for the least square solution $z_{LS}$ together with their relative error $\eta_i$ (dimensionless quantity) and variance $\nu_i$; 2- the solution components for the TSVD solution $z^{(k)}$ together with their relative error $\eta_i^{(k)}$ and variance $\nu_i^{(k)}$.

| | $x^*$ | $z^*$ | $z_{LS}$ | $\eta_i$ | $\nu_i$ | $z^{(k)}$ | $\eta_i^{(k)}$ | $\nu_i^{(k)}$ |
|---|---|---|---|---|---|---|---|---|
| $p_1$ | -6.984 | -0.070 | 3.715 | 54.190 | 19182.5715 | 0.004 | 1.052 | 0.0001 |
| $p_2$ | -1.114 | -0.011 | 0.342 | 31.715 | 23.2871 | 0.003 | 1.242 | 0.0005 |
| $p_3$ | -3.480 | -0.035 | -2.690 | 76.285 | 7384.9940 | 0.001 | 1.022 | 0.0000 |
| $p_4$ | -2.745 | -0.027 | 3.389 | 124.470 | 82380.0339 | -0.000 | 1.000 | 0.0000 |
| $p_5$ | 0.086 | 0.001 | 0.048 | 54.082 | 0.6575 | -0.004 | 5.139 | 0.0009 |
| $p_6$ | -8.776 | -0.088 | 67.445 | 769.477 | 1581516.7404 | 0.000 | 1.001 | 0.0000 |
| $p_7$ | -5.265 | -0.053 | -0.999 | 17.970 | 57.1478 | -0.005 | 0.900 | 0.0001 |
| $p_8$ | -6.640 | -0.066 | -0.344 | 4.176 | 7981.3944 | -0.016 | 0.757 | 0.0013 |
| $p_9$ | -10.292 | -0.103 | 133.504 | 1298.187 | 494620.7529 | -0.006 | 0.946 | 0.0002 |
| $p_{10}$ | -3.035 | -0.030 | -0.003 | 0.889 | 2.6980 | -0.011 | 0.632 | 0.0008 |
| $p_{11}$ | 3.539 | 0.035 | 1.075 | 29.370 | 79.5237 | -0.003 | 1.083 | 0.0003 |
| $p_{12}$ | 4.736 | 0.047 | 0.045 | 0.053 | 0.0256 | 0.044 | 0.080 | 0.0003 |
| $p_{13}$ | -0.772 | -0.008 | 1.042 | 135.879 | 8676.9499 | -0.028 | 2.616 | 0.0033 |
| $p_{14}$ | 3.261 | 0.033 | 0.072 | 1.196 | 4.3971 | 0.004 | 0.866 | 0.0001 |
| $p_{15}$ | 5.533 | 0.055 | 0.084 | 0.515 | 0.0548 | 0.058 | 0.053 | 0.0003 |
| $p_{16}$ | 4.082 | 0.041 | 0.092 | 1.265 | 1.2521 | 0.004 | 0.904 | 0.0001 |
| $p_{17}$ | 5.178 | 0.052 | 1.631 | 30.497 | 8160.2559 | 0.000 | 0.997 | 0.0005 |
| $C_{\text{lab}}$ | 6.237 | 0.062 | 1.002 | 15.073 | 8237.5716 | 0.019 | 0.697 | 0.0020 |
| $C_{\text{f}}$ | 4.073 | 0.041 | 1.348 | 32.090 | 8070.9246 | 0.005 | 0.888 | 0.0001 |
| $C_{\text{r}}$ | 6.858 | 0.069 | 1.788 | 25.067 | 8300.6094 | -0.012 | 1.170 | 0.0008 |
| $C_{\text{w}}$ | 8.341 | 0.083 | -318.175 | 3815.484 | 7436253.3370 | 0.000 | 0.999 | 0.0000 |
| $C_{\text{l}}$ | 5.961 | 0.060 | 0.568 | 8.532 | 8550.3479 | -0.006 | 1.097 | 0.0002 |
| $C_{\text{s}}$ | 8.956 | 0.090 | -134.334 | 1500.869 | 483025.4281 | -0.006 | 1.064 | 0.0002 |

In the next section we consider the concept of a resolution matrix, which allows for a finer analysis of the solution of the linear problem.

### 4.3 Resolution matrix

As suggested by equations (42) and (45), finding a solution $z$ amounts to constructing a *generalised inverse* $\mathbf{H}^g$ such that formally

$$z = \mathbf{H}^g d. \tag{47}$$

The generalised inverse is the operator representing any method, direct or iterative, used to solve the linear inverse problem, including or not any kind of regularisation. In the previous section we considered two examples of generalised inverse, the pseudo inverse and the truncated inverse obtained using TSVD. The generalised inverse can be used to define operators which directly address the conditions for well posedness for the linearised problem. Assuming a true state $z^*$ exists, possibly

unknown, then using equation (38) and (47) we can define an operator $\mathbf{N}$ called the *model resolution*

 *matrix* which relates the solution $z$ to the true state

$$z = \mathbf{H}^g \mathbf{H} z^* = \mathbf{N} z^*. \tag{48}$$

This matrix gives a practical tool to analyse the resolution power of an inverse method, that is its ability to retrieve the true state, including or not any regularisation method: the closer $\mathbf{N}$ is to the identity the better the resolution. Moreover the trace of the matrix defines a natural notion of *infor-*

 *mation content* (IC). Similarly a *data resolution matrix* can be defined to study how well data can be reconstructed and its diagonal elements naturally define a notion of *data importance*. For the two examples of generalised inverse presented in the previous section we obtain the following resolution matrices:

$$\mathbf{N} = \mathbf{H}^\dagger \mathbf{H}, \tag{49}$$

 for the pseudo-inverse and

$$\mathbf{N} = \mathbf{V}_k \mathbf{V}_k^T, \tag{50}$$

for the truncated pseudo inverse. In the first case the IC equals the number of non zero singular values, in the second case the IC equals the truncation rank $k$. An in-depth theoretical and practical analysis of these concepts and those introduced in the remainder of this section can be found in

 Menke (1984).

While the model resolution matrix allows us to see how a solution strategy maps the true state variables to the solution of the inverse problem, and to see how well and how independently the state variables can be recovered, one also needs to assess the uncertainty of the solution. This can be studied using the so called *unit covariance matrix*, $\mathbf{C}$, defined using the generalised inverse as

 $$\mathbf{C} = \mathbf{H}^{gT} \mathbf{H}^g. \tag{51}$$

By characterising the degree of error amplification that occurs in the mapping from the true state to the solution of the inverse problem, the unit covariance matrix is a crucial object to study the stability of the solution with respect to observational noise. The unit covariance matrix defined by (51) agrees with the covariance matrices given in the previous section by (43) for the pseudo-inverse,

 and by (46) when TSVD is applied.

### 4.4 Resolution for LAI operator

We now study the model resolution matrix for the LAI observation operator at Morgan Monroe State Forest. In the first instance we will demonstrate the resolution power of the LAI signal without regularisation using the pseudo-inverse as generalised inverse first, and then apply TSVD to show how

 using the truncated pseudo-inverse affects resolution. In a second case we will study the added value

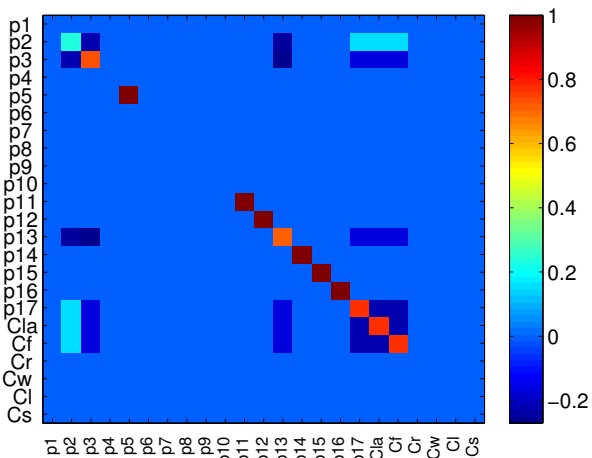

**Figure 6.** Model resolution matrix for the LAI operator.

of the EDCs in terms of resolution.

As previously, we linearise equation (28) about the point $x^*$ given in Table 2. The trace of the resolution matrix obtained using the pseudo-inverse as generalised inverse is 10, and this means that 10 independent variables can be estimated using LAI. These independent variables are not the variables in which the system is expressed but a linear transformation can be found to express the system in terms of the independent variables. Figure 6 shows the model resolution matrix for LAI. As shown in section 2.3 with the sensitivity analysis, 11 out of the 23 variables are not sensitive to LAI, and this can be seen in the resolution matrix by the diagonal terms which are zero, represented by the blue colour. In contrast the diagonal elements corresponding to sensitive variables have positive values, represented by colours ranging from light blue to red. Figure 6 also shows that whereas $p_5$, $p_{11}$, $p_{12}$, $p_{14}$, $p_{15}$ and $p_{16}$ are perfectly resolved (the corresponding elements are coloured in brown or dark red), there exist linear combinations between the remaining sensitive variables and this explains why only 10 independent variables can be estimated from the 12 sensitive variables.

For the study of the unit covariance matrix we restrict ourselves to the sensitive variables, this amounts to removing the columns corresponding to the non sensitive variables, containing only null elements, from the observability matrix. The dependency of the solution on observational noise can be studied by looking at Figure 7 where the diagonal elements of the unit covariance matrix, corresponding to the variance of each element of the solution obtained using the pseudo-inverse, are represented in log scale. Except for $p_5$, $p_{12}$, $p_{15}$ and $p_{16}$ all variances are shown to be large.

As previously, we illustrate a simple regularisation strategy, TSVD, and show its effects on both resolution and stability. Figure 8 shows the resolution matrix for LAI with optimal truncation rank $k = 6$. The IC decreases to 6. We see that whereas $p_5$, $p_{12}$, $p_{15}$ and $p_{16}$ remain almost perfectly resolved, $p_{13}$, $p_{17}$ and $C_{lab}$ are only partially resolved and the remaining variables are not resolved

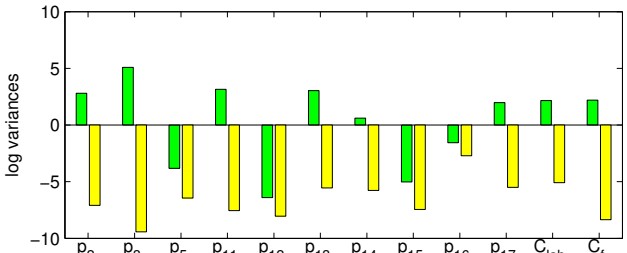

**Figure 7.** Diagonal elements (log scale) of the unit covariance matrix for the LAI operator using the pseudo inverse in green, using TSVD in yellow.

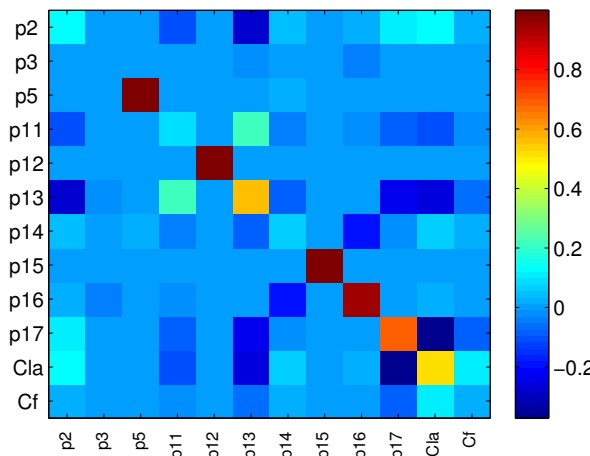

**Figure 8.** Model resolution matrix for the LAI operator using TSVD with truncation rank $k = 5$.

properly. Figure 7 shows the corresponding diagonal elements of the unit covariance matrix, we see
that the variances have been drastically reduced. This example shows how regularisation ensures stability at the price of losing resolution.

We now consider the effect of incorporating the static EDCs into the variational framework in terms of resolution. The static EDCs are given by the seven first EDCs, the linear problem is then given by

$$
\begin{bmatrix} \mathbf{H} \\ \tilde{\mathbf{G}} \end{bmatrix} \mathbf{z} = \begin{pmatrix} \mathbf{d} \\ \mathbf{f} \end{pmatrix},
\tag{52}
$$

where

$$
\tilde{\mathbf{G}} = g(\mathbf{x}^*)^{-1} \mathbf{\Sigma}^{-1/2} \mathbf{G}
\tag{53}
$$

with $\mathbf{\Sigma}^{-1/2}$ the inverse of the symmetric square root of the covariance matrix $\mathbf{\Sigma}$, defined in Section 3.3, restricted to the seven first components. The static EDCs depend only on 13 out of the 23

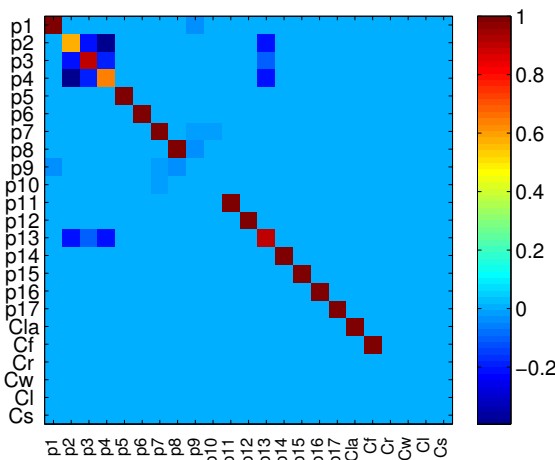

**Figure 9.** Model resolution matrix for LAI and static EDCs as defined by equation 52.

variables, namely $p_1$ to $p_{10}$, $p_{12}$, $p_{13}$ and $p_{15}$, this can be seen on the matrix $\mathbf{G}$ where the columns corresponding to the remaining variables are null. Together with LAI observations, whose sensitive variables are represented in Figure 2, we therefore have 19 sensitive variables. The model resolution matrix corresponding to the operator on the left hand side of equation (52), obtained using the pseudo inverse, is depicted in Figure 9. The trace of the model resolution matrix gives an IC of 16, 13 variables are perfectly resolved, 4 variables show linear dependencies ($p_2$, $p_3$, $p_4$ and $p_{13}$). However, although $p_9$ and $p_{10}$ are sensitive variables they do not appear to be resolved at all: inspecting the linear operator $\mathbf{G}$ shows that the non zero components corresponding to $p_9$ and $p_{10}$ are several order of magnitude smaller than the other components.

This example shows clearly the benefit of introducing the static EDCs to help estimating poorly constrained or otherwise undetermined components.

## 5 Experiments at Ameriflux sites

We now consider a real experiment at the Morgan Monroe state forest. At this Ameriflux site, 13 years of MODIS LAI monthly mean observations from 2001 to 2013, NEE, GPP and thus RESP observations from 2001 to 2005 are available. Our goal is to study two different aspects; the first one is the impact of using multiple data streams: how does it affect uncertainty of the predicted fluxes? how well do we predict non-observed fluxes? And the second one is to use the static EDCs and to assess their utility in constraining poorly sensitive variables.

**Table 3.** Experiment set up summary: in Exp 1 we use LAI and bounds constraints (BDS), in Exp 2 we use LAI, NEE and BDS and so on.

|  | LAI | NEE | GPP | RESP | BDS | EDCs |
|---|---|---|---|---|---|---|
| Exp 1 | • | | | | • | |
| Exp 2 | • | • | | | • | |
| Exp 3 | • | • | • | • | • | |
| Exp 4 | • | | | | • | • |
| Exp 5 | • | • | | | • | • |
| Exp 6 | • | • | • | • | • | • |

When including all terms the cost function, $J_{\text{TOT}}$, becomes

$$
\begin{aligned}
J_{\text{TOT}}(\boldsymbol{x}) = & \frac{\lambda_{\text{L}}}{2} \|\boldsymbol{h}_{\text{L}}(\boldsymbol{x}) - \boldsymbol{y}_{\text{L}}\|^2 + \frac{\lambda_{\text{N}}}{2} \|\boldsymbol{h}_{\text{N}}(\boldsymbol{x}) - \boldsymbol{y}_{\text{N}}\|^2 \\
& + \frac{\lambda_{\text{G}}}{2} \|\boldsymbol{h}_{\text{G}}(\boldsymbol{x}) - \boldsymbol{y}_{\text{G}}\|^2 + \frac{\lambda_{\text{R}}}{2} \|\boldsymbol{h}_{\text{R}}(\boldsymbol{x}) - \boldsymbol{y}_{\text{R}}\|^2 \\
& + \frac{\lambda_c}{2} \|\log(\boldsymbol{g}(\boldsymbol{x})) - \boldsymbol{c}\|^2_{\boldsymbol{\Sigma}} + \frac{\lambda_0}{2} \|\boldsymbol{x} - \boldsymbol{x}_0\|^2_{\mathbf{B}} \\
= & J_{\text{L}} + J_{\text{N}} + J_{\text{G}} + J_{\text{R}} + J_c + J_0,
\end{aligned}
$$

where subscripts L, N, G and R stand for LAI, NEE, GPP and RESP respectively. The vectors $\boldsymbol{y}_{\text{L}}$,

$\boldsymbol{y}_{\text{N}}$, $\boldsymbol{y}_{\text{G}}$ and $\boldsymbol{y}_{\text{R}}$ represent the observation vectors for LAI, NEE, GPP and RESP respectively. The scalars $\lambda_{\text{L}}$, $\lambda_{\text{N}}$, $\lambda_{\text{G}}$ and $\lambda_{\text{R}}$ take the value 0 or 1 depending on whether or not the corresponding data stream is included in the experiment. The scalar $\lambda_c$ takes the value 0 or 1 depending on whether we include the EDCs and $\lambda_0$ takes the value 1.

We perform six experiments summarised in Table 3. In experiment 1, Exp 1, we use only LAI

observations and bounds constraints so that in the cost function $J_{\text{TOT}}$ we set $\lambda_L = 1$ and $\lambda_0 = 1$, and the other $\lambda$s are set to zero. For Exp 2, we use LAI and NEE observations, that is we set $\lambda_L = 1$, $\lambda_N = 1$ and $\lambda_0 = 1$; the other $\lambda$s are set to zero. We proceed similarly for the remaining experiments. Here we assimilate all data streams simultaneously; it is not our intention to question what method best accommodates multiple data streams, MacBean et al. (2016) addresses this question using a

simple C-cycle model. Moreover we choose to assume the same statistical error for all data streams and set their error covariance matrix equal to the identity. To avoid being trapped at meaningless local minima, the experiments are performed multiple times using different initialisation parameter sets and results for the best candidate only are reported.

The results of the experiments are presented in Table 4 where each element of $J_{\text{TOT}}$ is given for

all experiments, and in Table 5 where the solution components and their variance are presented for all experiments. Results of Table 4 show that $J_{\text{L}}$ is the smallest in Exp 1 when LAI only is used. In Exp 2, when adding NEE we see that $J_N$ decreases from 109.012 in Exp 1 to 15.263, and $J_{\text{G}}$ slightly decreases as compared to Exp 1, but on the contrary $J_{\text{R}}$ increases. In Exp 3 we see that all costs drastically decrease compared to their initial values. Going from Exp 1 to Exp 3 $J_0$ slightly

increases, adding more data streams constrains more parameters and the parameters shift from their prior value which may cause $J_0$ to increase. Similar observations can be made for Exp 4 to Exp 6, moreover we see that including the EDCs only slightly affects the costs, a reason for this might be that EDCs help constraining the less sensitive parameters for which the costs are the less sensitive as suggested by the sensitivity analysis depicted in Figure 2. To see the effect of the EDCs we need to

look at Table 5, which details the solution components together with their relative variance defined by the ratio of the variance by the parameter range. In Exp 1 we see that the variables with the smallest relative variance are the most sensitive parameters as illustrated in Figure 2; they are $p_2$, $p_5$, $p_{10}$, $p_{12}$, $p_{14}$, $p_{15}$, $p_{16}$ and $p_{17}$. We recall that the sensitivity analysis of section 2.3 was performed by averaging sensitivities for an ensemble of initial parameter sets, therefore the ranking shown in

figure 2 may not be reflected in the relative variances. As we include NEE in Exp2 we see that most relative variances decrease, especially for $p_8$, $p_9$, $p_{10}$, $p_{13}$, $C_{\text{lab}}$, $C_{\text{f}}$ and $C_{\text{l}}$. The only variable whose relative variance increases is $p_{14}$, but as shown in figure 2 $p_{14}$ has very small sensitivity. In Exp 3 most relative variances decrease. The values are still large though for $p_1$, $p_3$, $p_4$, $p_6$, $p_9$, $C_{\text{r}}$ and $C_{\text{l}}$. Again similar features can be observed for Exp 4 to Exp 6 but a clear improvement can be

seen for most variables except for $C_{\text{r}}$ which is not constrained by the seven first EDCs. Finally, the last column of Table 4 shows the computation time for each experiments. As expected we see that the more observation streams we consider the longer the experiment takes to run, and incorporating the EDCs increases the computation time. However we stress that these figures are several orders of magnitude less than the time required to perform the same experiments using the current gold

standard MCMC approach used in Bloom and Williams (2015).

     Figures 10 and 11 show the predicted fluxes for LAI, NEE, GPP and RESP for the result of Exp 6. We can see a good agreement between modelled fluxes and observations. The uncertainty of the predicted fluxes is evaluated by modelling an ensemble of trajectories from a 95% ellipsoid of the posterior truncated Gaussian distribution: these trajectories are represented as grey curves on Figure

10 and 11. Figure 12 shows the posterior parameter distribution marginals for $p_1$, $p_7$, $p_{16}$, and $C_{\text{f}}$ for Exp 6, illustrating the four different cases where: as for $p_{16}$ most of the marginal is contained in the parameter range; the marginal is truncated on the left or the right as for $p_7$ or $C_{\text{f}}$ and truncated on both sides for $p_1$.

## 6   DALEC-SP

In the previous section we used EDCs within 4DVAR and showed their benefit in reducing drastically the uncertainty of otherwise undetermined variables. However we only included the static EDCs which do not require a model run: including more EDCs often leads to convergence issues, the solution and its uncertainty become subject to caution.

     As shown in Chuter et al. (2015) for the previous DALEC evergreen and deciduous models, the

**Table 4.** Costs for the results of the inverse modelling experiments. The last column reports the computation time in seconds for the experiment.

| | $J_L$ | $J_N$ | $J_G$ | $J_R$ | $J_0$ | $J_c$ | time (s) |
|---|---|---|---|---|---|---|---|
| $x_{init}$ | 179.525 | 353.229 | 1265.556 | 419.696 | 0.003 | 7.157 | 0.000 |
| Exp 1 | 14.083 | 109.012 | 153.475 | 45.415 | 0.017 | 2.498 | 2.722 |
| Exp 2 | 19.188 | 15.263 | 145.349 | 131.963 | 0.018 | 3.704 | 7.541 |
| Exp 3 | 25.089 | 16.737 | 36.155 | 17.842 | 0.020 | 4.643 | 5.886 |
| Exp 4 | 14.083 | 107.420 | 152.908 | 45.480 | 0.016 | 2.498 | 5.012 |
| Exp 5 | 19.193 | 15.262 | 145.254 | 131.878 | 0.018 | 3.701 | 9.045 |
| Exp 6 | 25.059 | 16.699 | 36.143 | 17.826 | 0.019 | 4.642 | 8.215 |

**Table 5.** Results of the inverse modelling experiments. The solution components together with their relative variance, in bracket, are given for each experiment. The vector $x_{init}$ is the randomly chosen parameter set satisfying the EDCs that initialises the minimisation routine.

| | $x_{init}$ | Exp 1 | Exp 2 | Exp 3 | Exp 4 | Exp 5 | Exp 6 |
|---|---|---|---|---|---|---|---|
| $p_1$ | -5.172 | -8.059 (1.727) | -8.248 (1.471) | -8.282 (1.021) | -6.323 (0.125) | -6.486 (0.095) | -6.415 (0.087) |
| $p_2$ | -0.947 | -0.885 (0.207) | -1.106 (0.120) | -1.085 (0.030) | -0.770 (0.126) | -0.960 (0.093) | -0.984 (0.019) |
| $p_3$ | -4.318 | -2.673 (0.955) | -2.944 (0.849) | -3.376 (0.894) | -3.001 (0.950) | -4.419 (0.965) | -3.461 (0.880) |
| $p_4$ | -1.493 | -2.649 (0.978) | -2.813 (0.961) | -2.692 (0.936) | -1.308 (0.111) | -1.703 (0.142) | -1.354 (0.097) |
| $p_5$ | 1.123 | 0.117 (0.003) | 0.153 (0.002) | 0.085 (0.000) | 0.149 (0.002) | 0.172 (0.001) | 0.091 (0.000) |
| $p_6$ | -7.959 | -8.752 (0.922) | -8.870 (0.911) | -8.707 (0.919) | -7.151 (0.170) | -7.039 (0.108) | -7.681 (0.458) |
| $p_7$ | -7.432 | -6.908 (1.151) | -6.373 (0.941) | -5.064 (0.224) | -7.236 (0.221) | -5.961 (0.109) | -6.144 (0.108) |
| $p_8$ | -5.281 | -6.908 (1.151) | -6.906 (0.316) | -6.522 (0.078) | -5.565 (0.113) | -5.753 (0.052) | -6.160 (0.039) |
| $p_9$ | -16.012 | -11.513 (2.303) | -10.075 (0.995) | -11.411 (1.514) | -14.592 (2.303) | -8.384 (0.036) | -11.266 (0.143) |
| $p_{10}$ | -3.041 | -3.272 (0.373) | -3.296 (0.085) | -3.036 (0.055) | -3.244 (0.373) | -3.045 (0.072) | -3.079 (0.055) |
| $p_{11}$ | 2.792 | 3.829 (0.540) | 4.026 (0.163) | 3.542 (0.003) | 3.630 (0.286) | 4.198 (0.232) | 3.544 (0.003) |
| $p_{12}$ | 3.549 | 4.626 (0.002) | 4.739 (0.000) | 4.735 (0.000) | 4.627 (0.002) | 4.733 (0.000) | 4.735 (0.000) |
| $p_{13}$ | -1.768 | -0.693 (0.130) | -0.996 (0.067) | -0.795 (0.046) | -0.779 (0.039) | -1.193 (0.037) | -0.801 (0.029) |
| $p_{14}$ | 3.343 | 4.013 (0.034) | 3.291 (0.123) | 3.292 (0.052) | 3.940 (0.029) | 3.526 (0.048) | 3.247 (0.079) |
| $p_{15}$ | 5.656 | 5.512 (0.001) | 5.528 (0.001) | 5.531 (0.000) | 5.521 (0.001) | 5.540 (0.000) | 5.534 (0.000) |
| $p_{16}$ | 4.529 | 4.115 (0.068) | 3.993 (0.025) | 4.095 (0.011) | 4.003 (0.061) | 4.071 (0.019) | 4.082 (0.010) |
| $p_{17}$ | 5.351 | 5.289 (0.213) | 5.278 (0.198) | 5.138 (0.165) | 5.142 (0.135) | 5.032 (0.110) | 5.098 (0.098) |
| $C_{lab}$ | 3.979 | 5.950 (0.115) | 6.031 (0.059) | 6.187 (0.040) | 5.884 (0.078) | 5.841 (0.039) | 6.151 (0.025) |
| $C_f$ | 5.389 | 4.677 (0.282) | 4.868 (0.068) | 4.038 (0.066) | 4.421 (0.302) | 4.342 (0.093) | 4.018 (0.051) |
| $C_w$ | 7.045 | 5.298 (1.151) | 5.829 (0.900) | 6.520 (0.096) | 5.309 (1.022) | 6.910 (0.226) | 7.061 (0.135) |
| $C_r$ | 9.753 | 8.406 (1.554) | 8.188 (1.533) | 8.318 (1.544) | 8.687 (1.554) | 6.771 (1.030) | 8.538 (1.415) |
| $C_l$ | 3.992 | 5.298 (1.151) | 7.307 (0.300) | 6.226 (0.161) | 5.963 (0.990) | 5.670 (0.263) | 5.985 (0.088) |
| $C_s$ | 9.721 | 8.406 (1.900) | 9.546 (1.188) | 8.603 (1.633) | 8.895 (1.900) | 7.140 (0.561) | 9.003 (0.718) |

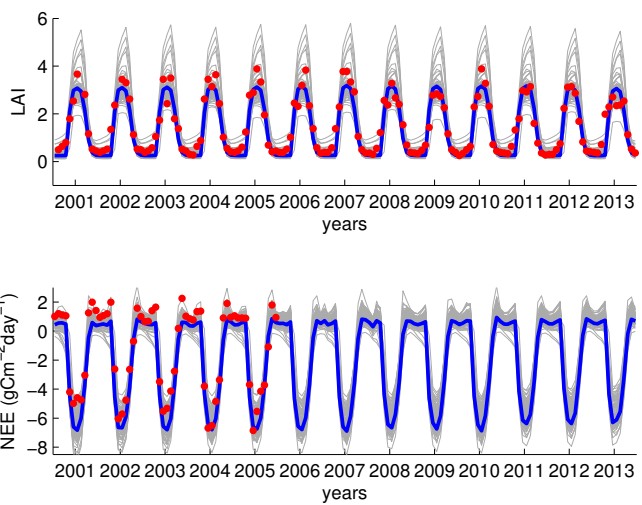

**Figure 10.** DALECv2 monthly estimates for LAI and NEE at Morgan Monroe State forest. The red dots are the observations, the blue trajectories are obtained using the 4DVAR analysis, the grey trajectories are ensemble runs obtained from a 95% confidence sample of the posterior pdf.

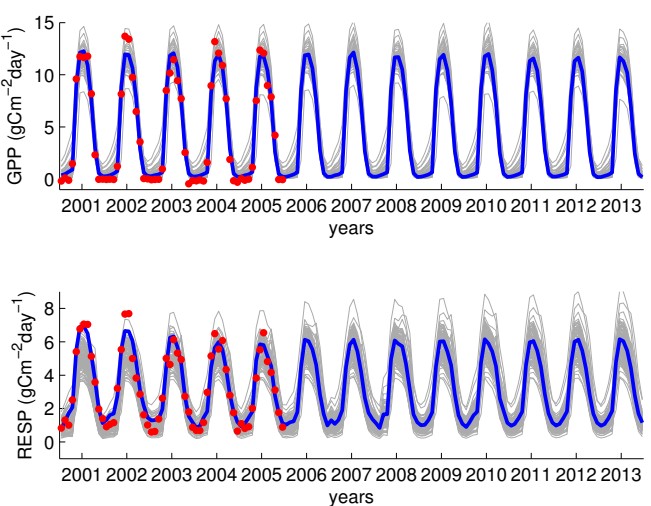

**Figure 11.** DALECv2 monthly estimates for GPP and RESP at Morgan Monroe State forest. The red dots are the observations, the blue trajectories are obtained using the 4DVAR analysis, the grey trajectories are ensemble runs obtained from a 95% confidence sample of the posterior pdf.

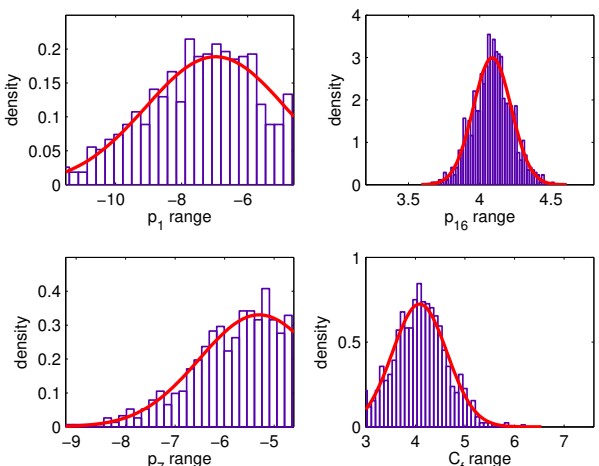

**Figure 12.** Posterior parameter distributions for parameters $p_1$, $p_7$, $p_{16}$ and $C_f$ for Exp 6. For each plot the limits of the abscissa correspond to the parameter range. The red curve is the Gaussian posterior distribution and the blue bars represent the sample used to produce the grey trajectories in figures 10 and 11.

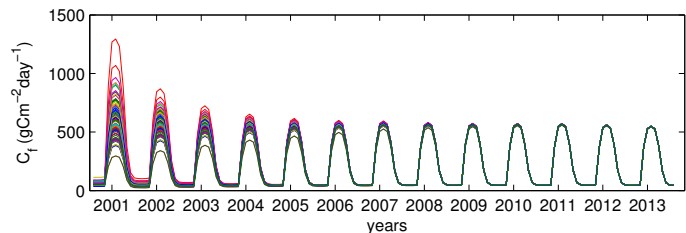

**Figure 13.** Pseudo periodical seasonal cycle for DALECv2. Using a given set of parameters and initial values for $C_w$ and $C_s$, 100 DALECv2 runs are performed using random initial values for $C_{lab}$, $C_f$, $C_r$, $C_l$. The plot shows the 100 trajectories for $C_f$.

545 evolution of the carbon pools for DALECv2 show a tipping point which depends on the parameters $p_1$ to $p_{17}$. Given a set of parameters **p** the fast carbon pools $C_{lab}$, $C_f$, $C_r$ and $C_l$ grow or decay rapidly to an equilibrium state. This equilibrium is either zero, the forest dies out, or a pseudo-periodical seasonal cycle as shown on Figure 13 for $C_f$. Moreover there exists a limit value below which any initial condition leads to the zero equilibrium and above which the equilibrium is a strictly positive

550 pseudo-periodical seasonal cycle.

 Here we consider ecosystems with no recent major disturbance, where the fast carbon pools are expected to be close to their pseudo-periodical cycle. To model these ecosystems, one can either restrict the parameter space by using the dynamic EDCs, or we can introduce a spin-up period during which the carbon pools reach their attractor. Given parameters $p_1$ to $p_{17}$ and initial values for $C_w$

555 and $C_s$ a first run of DALECv2 is performed to obtain a state which is closer to a pseudo-periodical

**Table 6.** Results of the inverse modelling experiments with DALEC-SP model showing the log of the initial carbon pools (gCm$^{-2}$). The rows labelled (SP) correspond to the solution components for DALEC-SP, the other rows reproduce the results for DALEC as reported in Table 5. As previously the relative variances are given in bracket.

| | $x_{\text{init}}$ | Exp 1 | Exp 2 | Exp 3 | Exp 4 | Exp 5 | Exp 6 |
|---|---|---|---|---|---|---|---|
| $C_{\text{lab}}$ | 3.979 | 5.950 (0.115) | 6.031 (0.059) | 6.187 (0.040) | 5.884 (0.078) | 5.841 (0.039) | 6.151 (0.025) |
| $C_{\text{lab}}$ (SP) | NA | 5.461 (0.106) | 5.961 (0.015) | 6.064 (0.008) | 5.097 (1.572) | 5.914 (0.074) | 6.048 (0.011) |
| $C_{\text{f}}$ | 5.389 | 4.677 (0.282) | 4.868 (0.068) | 4.038 (0.066) | 4.421 (0.302) | 4.342 (0.093) | 4.018 (0.051) |
| $C_{\text{f}}$ (SP) | NA | 4.358 (0.112) | 4.602 (0.034) | 3.966 (0.029) | 4.102 (0.633) | 4.586 (0.079) | 4.024 (0.043) |
| $C_{\text{r}}$ | 7.045 | 5.298 (1.151) | 5.829 (0.900) | 6.520 (0.096) | 5.309 (1.022) | 6.910 (0.226) | 7.061 (0.135) |
| $C_{\text{r}}$ (SP) | NA | 4.702 (0.765) | 5.069 (0.575) | 5.072 (0.665) | 6.078 (0.260) | 5.323 (0.316) | 6.398 (0.116) |
| $C_{\text{w}}$ | 9.753 | 8.406 (1.554) | 8.188 (1.533) | 8.318 (1.544) | 8.687 (1.554) | 6.771 (1.030) | 8.538 (1.415) |
| $C_{\text{w}}$ (SP) | 7.201 | 5.298 (0.853) | 5.289 (0.852) | 5.285 (0.853) | 5.298 (0.853) | 5.320 (0.852) | 5.203 (0.853) |
| $C_{\text{l}}$ | 3.992 | 5.298 (1.151) | 7.307 (0.300) | 6.226 (0.161) | 5.963 (0.990) | 5.670 (0.263) | 5.985 (0.088) |
| $C_{\text{l}}$ (SP) | NA | 6.361 (0.276) | 6.590 (0.215) | 6.791 (0.181) | 5.412 (0.226) | 5.928 (0.074) | 6.056 (0.033) |
| $C_{\text{s}}$ | 9.721 | 8.406 (1.900) | 9.546 (1.188) | 8.603 (1.633) | 8.895 (1.900) | 7.140 (0.561) | 9.003 (0.718) |
| $C_{\text{s}}$ (SP) | 9.793 | 8.406 (1.900) | 8.550 (0.799) | 9.324 (1.359) | 8.406 (1.900) | 8.940 (0.222) | 10.453 (1.090) |

cycle for the fast carbon pools. The steady state trajectories are then used to initialise the fast carbon pools. For this DALECv2-"spin-up" model, DALEC-SP, the state variable is therefore formed of the seventeen parameters $p_1, ..., p_{17}$ and the initial conditions for $C_{\text{w}}$ and $C_{\text{s}}$.

DALEC-SP offers several advantages: some of the EDCs such as those controlling the growth and the half life period of carbon pools are almost automatically satisfied; this reduces largely the time required to generate the pdf $p(\boldsymbol{c}|\boldsymbol{x})$. Moreover as the sensitivity analysis and the resolution matrices showed, the fast carbon pools are variables that are not highly sensitive to the signals that we observe, and therefore reducing the number of variables by removing the fast carbon pools is likely to improve the overall conditioning of the inverse problem. To investigate this assertion we perform Exp 1 to 6 using DALEC-SP. The solution components and their variance for the carbon pools are presented in Table 6; the results for the parameters $p_1$ to $p_{17}$ are not reported as they do not significantly differ from what was observed and reported in Table 5 for DALECv2. For the fast carbon pools, which are not directly estimated during the assimilation process, we start by taking a sample of the posterior pdf and then we run DALEC-SP for this sample. The values presented in Table 6 represent the means and variances of the fast carbon pools after the spin up period. Except for two anomalies in Exp 4, where the uncertainty for $C_{\text{lab}}$ and $C_{\text{f}}$ is larger with DALEC-SP as compared to DALECv2, almost all uncertainties for all experiments are smaller with DALEC-SP. Despite some improvement in Exp 2, 4 and 5, the uncertainty for $C_{\text{s}}$ is still large.

## 7 Discussion

To our knowledge, this paper presents the first application of variational methods for an inverse modelling experiment using DALEC. Over the last fifteen years many studies have validated the use of DALEC together with various types of data streams to infer ecological parameters at site level but ensemble Kalman filter first and then Monte Carlo methods were privileged. At the same time 4DVAR was successfully used at global scale to constrain ecosystem parameters in carbon

cycle data assimilation system (CCDAS). In Rayner et al. (2005) the Biosphere Energy Transfer Hydrology model (BETHY) is coupled with the transport model TM2, and satellite observations of photo-synthetically active radiation and atmospheric $CO_2$ concentration observations are used to optimise model parameters. In this context Kemp et al. (2014) investigated how to constrain the 4DVAR problem in CCDAS through a number of different methods: using constrained optimisation,

adding a penalty term and applying parameter transformations. They concluded that using parameter transformations give the best results. In our context the three methods were investigated: Gaussian anamorphosis where priors based on the distribution of parameters satisfying the EDCs were considered, constrained optimisation as stated in section 3.3, and adding a penalty term to account for the EDCs. The latter solution which is the main interest of this publication was found to be the most

successful in our case.

The complexity of global scale experiments still limit the application of fully nonlinear methods such as MCMC. In Ziehn et al. (2012) a comparison between the MCMC Metropolis-Hastings approach and 4DVAR for the BETHY-TM2 CCDAS framework is performed. This study reports a computation time of less than one hour for the variational method and about height months for the overall

MCMC computation. For our setting, DALECv2 site based experiment, the complexity is relatively small and a MCMC approach is affordable. Used in Bloom and Williams (2015), the MCMC approach for DALEC is studied in detail in Safta et al. (2015), the resulting parameter distributions suggest that 4DVAR and the inherent Gaussian approximation provides a reasonable posterior distribution.

As most variational methods, the analysis and application presented in this paper rely heavily on the possibility to derive the tangent linear model and its adjoint. DALECv2 was designed to take into account this requirement, in particular replacing the phenology process of the DALEC deciduous model was suggested in order to obtain differentiable processes. The model resolution matrix and the gradient of the cost function, including the additional term encoding the EDCs, are computed us-

ing adjoint techniques. Despite the increasing capacities offered by automatic differentiation tools, deriving and maintaining an adjoint code can be a complicated task, and, besides its limiting hypothesis, this is certainly one of the main reason for choosing alternatives to 4DVAR. In a paper in preparation we use ensemble methods to approximate the gradient of the cost function and to derive approximate resolution matrices, and the experiments presented in this paper are reproduced. The

approach, which no longer requires the adjoint, shows very promising results: firstly in terms of es-

timating parameters, and secondly in terms of computation time by using graphic processing units (GPU) to perform massive parallel computations.

Designing a global scale experiment involving a coupling between DALEC and a transport model has been considered but is still at an early stage. As presented in Bloom and Williams (2015), the EDCs were originally introduced to constrain unresolved parameters at site level where, in the absence of any other information, only MODIS LAI observations were available. In theory there is no restriction to readily apply the same constraints at global scale however their efficiency highly depends on the nature of the coupling between the ecosystem model and the transport model and on the observation streams considered. Nonetheless in this context 4DVAR remains the only reasonable method to consider in terms of computer resources, and our study demonstrate that the current research efforts to develop regularisation strategies fit well into the variational framework.

## 8 Conclusions

We used DALECv2 and combined multiple data streams - MODIS monthly LAI and monthly NEE, GPP and RESP at an Ameriflux site - together with ecological constraints to estimate model parameters and initial conditions and to provide uncertainty characterisation for predicted fluxes. DALECv2 is a simple model; it represents the basic processes at the heart of more sophisticated models of the carbon cycle and, besides its large modelling skills, its simplicity allows for close mathematical scrutiny. Here we adopted a variational approach where the tangent linear model and its adjoint play a major role in 1) facilitating a linear analysis which allows to understand the nature of the ill-posed problem and to evaluate strategies to regularise it; 2) finding a posterior distribution for the state variables.

We performed a sensitivity analysis using a direct method that consists in studying the first order derivatives of the output computed using an adjoint method. The sensitivity analysis is a prerequisite to any work with a model, but there is a paucity of literature on this topic in connection with DALEC. Our analysis reveals generic issues that will be encountered in many inverse modelling strategies. Studying the first order inverse problem, we discussed how noise affects the stability of the solution and we illustrated a simple regularisation method. We then introduced the notion of model resolution matrix and showed how this can be used to diagnose the ill-posedness of an inverse problem and evaluate the result of regularisation strategies. While some of our findings may be anticipated in the framework of a simple model, it is important to describe these tools and their interpretation as similar analyses can be readily applied to a wide range of more complex models.

Bloom and Williams (2015) proved the benefit of the EDCs in constraining poorly resolved components of the carbon cycle and recommended their use for inverse modelling problems. We successfully incorporated the EDCs within the context of variational data assimilation. Our results confirm that the EDCs regularise an otherwise ill-posed problem and efficiently reduce the uncertainty of

predicted fluxes, and thus comfort the recommendation of Bloom and Williams (2015). Moreover, our modification to DALECv2, DALEC-SP which includes a spin-up period, offers an alternative to some EDCs that facilitates the variational approach.

This study did not aim at providing an exhaustive account for the capability of variational tools, nor at exploring all aspects of the EDCs for the inverse problem for DALEC. The objectives were to use 4DVAR and show that it offers a suitable framework to solve efficiently, robustly and quickly the inverse problem for DALEC, and to present some methodology to analyse some issues that affect most methods based on Bayesian inference.

## 9 Code availability

The model and inversion code, together with the drivers, observational data and experiment results are available at: https://zenodo.org/record/269937.

*Acknowledgements.* The authors would like to thank the anonymous referees for their valuable comments which helped to improve the manuscript. This project was funded by the NERC National Centre for Earth Observation, NCEO, UK. We acknowledge US-MMS AmeriFlux site for its data records. Research at the MMSF site was supported by the Office of Science (BER), U.S. Department of Energy, Grant No. DE-FG02-07ER64371. AmeriFlux is funded by the United States Department of Energy (DOE – TES), Department of Commerce (DOC – NOAA), the Department of Agriculture (USDA – Forest Service), the National Aeronautics and Space Administration (NASA), the National Science Foundation (NSF). We are grateful to J. Exbrayat and A. Bloom for providing us with meteorological drivers, MODIS LAI observations and DALECv2 code. Finally we are grateful to M. Williams, J. Exbrayat, A. Bloom and T. Hill for comments and useful discussions.

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
