# Peer review of "Constraining DALEC v2 using multiple data streams and ecological constraints: analysis and application."

_Geoscientific Model Development, 2017_

## Referee Comment (RC1) · Anonymous Referee #1 · 18 Apr 2017

Review: "Constraining DALEC v2 using multiple data streams and ecological constraints: analysis and application" by Sylvain Delahaies et al.

Summary:

The manuscript by Delahaies et al. describes how to incorporate ecological and dynamic constraints (EDCs) into a 4DVar framework in order to estimate 23 parameters and initial conditions in a simple box model. Further, they show how the constraints can help to estimate otherwise underdetermined components. In their study, multiple data streams are combined, using observations from an Ameriflux site for LAI, NEE, GPP and respiration. Sensitivity analysis is performed to identify the most important parameters. They also introduce the resolutions matrix to diagnose the ill-posedness

of the linearised inverse problem.

The manuscript is well written and the details of the methods are described adequately. However, I am missing a broader discussion and some comments are made below. Nevertheless, I would recommend publications after minor revisions.

Comments:

It has already been shown in previous work that EDCs can be used as a form of regularisation to reject unrealistic parameter combinations in the same simple model. The novel aspect of the study by Delahaies et al. is that they include theses constraints directly in a 4DVar framework, which is really useful.

The 4DVar framework has already successfully been used to constrain terrestrial ecosystem parameters at the global scale in the Carbon Cycle Data Assimilation System (CCDAS) (i.e. Rayner at al., 2005). Kemp et al. (2014) also investigated how to constrain the 4DVar problem in CCDAS through a number of different methods including a constrained optimiser and parameter transformations. This should be included in the discussion.

I would also encourage the authors to comment on the feasibility of applying their framework at the global scale. It is great to see how the additional constraints work at the site level using a simple box model, but ultimately we would like to apply this to more complex global models. The authors argue that 4DVar is much faster than Monte Carlo methods, but it relies on the availability of adjoint code, which is not always easy to generate (i.e. model is not differentiable, etc.) .

Minor and technical comments: P3, L78: $CO_2$

P4, Table 1: Where to you get the range for the parameters and initial conditions from? Some of them seem to be different to what is stated in Bloom and Williams (2014).

P5, L128-129: You mention that you have 23 inequalities denoted EDC1 to EDC22. That's only 22? Can you provide the EDCS in the same way you provided EDC23 to

EDC25?

P8-P9, L208-211: How good is the Gaussian approximation of the posterior uncertainty? I guess that depends on the non-linearity of the model. You are using simple model and a Monte Carlo simulation should be possible (i.e. just sampling the cost function minimum). This kind of comparison has been done, for example, in Ziehn et al. (2012).

P9, L219 and throughout the manuscript: Your refer to Table 2.1, but this should probably be Table 2?

P10, L243: You refer to EDCs 4 and 6, but the reader does not know what they are. Can you please provide a complete list of the EDCs you are using?

P18, L439: Table 3 instead of Table 5?

P19, L449 Table 4 instead of Table 5? Can you please check references to tables throughout the manuscript?

References:

Rayner, P. J., M. Scholze, W. Knorr, T. Kaminski, R. Giering, and H. Widmann (2005), Two decades of terrestrial carbon fluxes from a carbon cycle data assimilation system (CCDAS), Global Biogeochem. Cycles, 19, GB2026, doi:10.1029/2004GB002254.

Kemp, S., Scholze, M., Ziehn, T., and Kaminski, T.: Limiting the parameter space in the Carbon Cycle Data Assimilation System (CCDAS), Geosci. Model Dev., 7, 1609-1619, doi:10.5194/gmd-7-1609-2014, 2014.

Bloom, A. A. and Williams, M.: Constraining ecosystem carbon dynamics in a data-limited world: integrating ecological "common sense" in a model–data fusion framework, Biogeosciences, 12, 1299-1315, doi:10.5194/bg-12-1299-2015, 2015.

Ziehn, T., M. Scholze, and W. Knorr (2012), On the capability of Monte Carlo and adjoint inversion techniques to derive posterior parameter uncertainties in terrestrial ecosys-

tem models, Global Biogeochem. Cycles, 26, GB3025, doi:10.1029/2011GB004185.

---

## Referee Comment (RC2) · Anonymous Referee #2 · 21 Apr 2017

This manuscript presents the application of an optimization method that uses multiple data-streams and constraints in the optimization of the DALEC ecosystem model. The topic of the manuscript is interesting and most likely it will be a good contribution.

I had two main issues with the current version of the manuscript. First, I have problems understanding the notation used throughout the ms. Second, it was unclear to me why the authors needed to optimize for the initial values of the model. I will elaborate better below.

As far as I know, DALEC can be expressed mathematically as a linear system of first

order differential equations of the form

$$\frac{dx}{dt} = u + A \cdot x \quad \text{with} \quad x(t=0) = x_0$$

where $x$ is a vector of state variables, $u$ a vector of C inputs to the ecosystem, and $A$ is a matrix containing cycling (turnover) rates in the diagonal, and transfer coefficients among pools in the off-diagonal entries. Linear autonomous systems of this type simply go to a steady-state value $x^*$ independent on the initial conditions $x_0$ according to

$$x^* = -A^{-1} \cdot u.$$

Without any environmental effects, this model should simply go to this steady-state value, but with environmental perturbations the system should stay in the vicinity of the steady-state, also independent on the initial conditions. It is therefore unclear to me, why do you have to optimize for the initial conditions of the model? Are you using a nonlinear version of DALEC? The discussion starting at line 289 seems to indicate this, but there is not a clear description of the model that clarify what type of nonlinear behavior is included in the version of DALEC used here.

I was also confused by the mix of terms: parameters, state-variables, variables, etc. throughout the manuscript. Due to this ambiguity, I had a hard time understanding sections 3.1 and 3.2. I also had problems with the term $h$, which was described as a vector of model output in line 142, and as a map in equation (2). Please clarify.

———————————————————

---

## Author Comment (AC1) · 18 May 2017

We thank referee 1 for his thorough reading of the manuscript and for useful comments. In our response below we have addressed all comments. Together with this response we submit a revision of the manuscript which accounts for the changes decsribed here.

**1   Tables references**

All tables references have been checked and corrected.

[Figure]

**2 Range of parameters**

The range of parameters in Table 1 have been corrected, they correspond to what is used in Bloom and Williams (2015).

**3 Ecological constraints**

Our implementation of the EDCs leads to a set of 29 inequalities denoted $EDC_1$ to $EDC_{29}$, we omitted to take into account the carbon pools growth constraints in the original manuscript.

In the original manuscript we chose to provide only a heuristic description of the inequalities, the complete description of which can be found in Bloom and Williams (2015). We thought that although justified for the sake of self consistency, detailing the EDCs did not bring any insight into the question addressed here and increased significantly unnecessary mathematical notation. Nonetheless we acknowledge the comment made by referee 1 and we have made substantial changes to section 2.2 to incorporate a complete description of the EDCs, running from line 123 to line 180 in the revised manuscript.

**4 Other comments**

A discussion section has been added to the manuscript to address the remaining comments.

Global scale experiments are discussed in reference to Rayner et al. (2005) and CC-DAS. The work of Kemp et al. (2014), which directly relates to what we discuss in our paper, is also cited. Although this preliminary work is not reported in our manuscript

the three approaches presented in Kemp et al. (2014) were evaluated for the preparation of the manuscript, and our conclusion was different: in our case incorporating the EDCs by adding a penalty term to the cost function was the most successful approach to constrain unresolved parameters and, most importantly in our case, to allow for a better uncertainty quantification.

We also refer to Ziehn et al. (2012) for their comparison between MCMC method and 4DVAR in CCDAS. A MCMC method is used in Bloom and Williams (2015), and in Safta et al. (2015), added in reference to the revised manuscript, a detailed analysis of MCMC for DALEC is performed. A comparison between MCMC and 4DVAR was beyond the scope of this paper, our intention was rather to establish 4DVAR as a suitable method for DALEC and the EDCs. Nonetheless a comparison between 4DVAR and fully non-linear methods is necessary, it is one of the aspects of our current work.

Finally we mention our current work on a hydrid ensemble-variational method. This approach provides an adjoint-free formulation of the variational problem and show promising results in the context discussed in the manuscript. This work is part of a paper in preparation.

**References**

Bloom, A. A. and Williams, M.: Constraining ecosystem carbon dynamics in a data-limited world: integrating ecological "common sense" in a model–data fusion framework, Biogeosciences, 12, 1299–1315, doi:10.5194/bg-12-1299-2015, 2015.

Kemp, S., Scholze, M., Ziehn, T., and Kaminski, T.: Limiting the parameter space in the Carbon Cycle Data Assimilation System (CCDAS), Geoscientific Model Development, 7, 1609–1619, doi:10.5194/gmd-7-1609-2014, http://www.geosci-model-dev.net/7/1609/2014/, 2014.

Rayner, P. J., Scholze, M., Knorr, W., Kaminski, T., Giering, R., and Widmann, H.: Two decades of terrestrial carbon fluxes from a carbon cycle data assimilation system (CCDAS), Global Biogeochemical Cycles, 19, n/a–n/a, doi:10.1029/2004GB002254, http://dx.doi.org/10.1029/2004GB002254, gB2026, 2005.

Safta, C., Ricciuto, D. M., Sargsyan, K., Debusschere, B., Najm, H. N., Williams, M., and

Thornton, P. E.: Global sensitivity analysis, probabilistic calibration, and predictive assessment for the data assimilation linked ecosystem carbon model, Geoscientific Model Development, 8, 1899–1918, doi:10.5194/gmd-8-1899-2015, http://www.geosci-model-dev.net/8/1899/2015/, 2015.

Ziehn, T., Scholze, M., and Knorr, W.: On the capability of Monte Carlo and adjoint inversion techniques to derive posterior parameter uncertainties in terrestrial ecosystem models, Global Biogeochemical Cycles, 26, n/a–n/a, doi:10.1029/2011GB004185, http://dx.doi.org/10.1029/2011GB004185, gB3025, 2012.

---

## Author Comment (AC2) · 18 May 2017

We thank referee 2 for his thorough reading of the manuscript and for useful comments. We have addressed all comments and tried our best to clarify the manuscript. Our response is described below. Together with this response we submit a revision of the manuscript which accounts for the changes decsribed here.

The version of DALEC used for this study, referred to as DALECv2 and described in details in Bloom and Williams (2015), is a nonlinear dynamical system, it is not a linear autonomous system. The trajectories of the carbon pools $\mathbf{C}$ are computed using the

recursion formula
$$\mathbf{C}^{t+1} = \mathbf{C}^t + \mathbf{f}(\mathbf{C}^t, \mathbf{p}, \phi(t))\Delta t, \tag{1}$$
where $\mathbf{f}$ is a nonlinear vector valued function of the carbon pools, the parameters $\mathbf{p}$ and the meteorological drivers $\phi(t)$, $\Delta t$ denoting the step time in month in our case. The nonlinear nature of the model is stressed out in the revised manuscript at lines 77 and 83. Moreover, the definition of $\mathbf{h}$ is clarified at line 197.

The main focus of the paper is on the vector $\mathbf{x} = \log([\mathbf{p}, \mathbf{C}_0])^T$. In section 2.3 first where we investigate the sensitivity of different outputs with respect to $\mathbf{x}$ and its components, and then in subsequent sections where $\mathbf{x}$ is estimated using inverse methods. The vector $\mathbf{x}$, denoting fixed quantities as initial conditions and parameters for the dynamical system DALECv2, is seen as the variable from the point of view of sensitivity analysis and inverse modelling and therefore its components are referred to as state variables, input variables or parameters interchangeably throughout the manuscript. This choice of terminology, stressed out at line 228, have been reinforced in the revised manuscript by adding the present paragraph at line 103.

**References**

Bloom, A. A. and Williams, M.: Constraining ecosystem carbon dynamics in a data-limited world: integrating ecological "common sense" in a model–data fusion framework, Biogeosciences, 12, 1299–1315, doi:10.5194/bg-12-1299-2015, 2015.

---

## Author Response (AR1)

**Response to reviewers for the manuscript: Constraining DALEC v2 using multiple data streams and ecological constraints.**

Sylvain Delahaies[1], Ian Roulstone[1], and Nancy Nichols[2]

[1]Department of Mathematics, University of Surrey, Guildford, UK.
[2]Department of Mathematics, University of Reading, Reading, UK.

*Correspondence to:* S. Delahaies (s.b.delahaies@surrey.ac.uk)

We thank both referees for their thorough reading of the manuscript and for useful comments. We have addressed all comments and tried our best to clarify the manuscript. Our responses to both referees are described in the follwing sections. Together with this response we submit a revision of the manuscript which accounts for the changes decsribed below.

**1 Response to referee 1**

**1.1 Tables references**

All tables references have been checked and corrected.

**1.2 Range of parameters**

The range of parameters in Table 1 have been corrected, they correspond to what is used in Bloom and Williams (2015).

**1.3 Ecological constraints**

Our implementation of the EDCs leads to a set of 29 inequalities denoted $EDC_1$ to $EDC_{29}$, we omitted to take into account the carbon pools growth constraints in the original manuscript.

In the original manuscript we chose to provide only a heuristic description of the inequalities, the complete description of which can be found in Bloom and Williams (2015). We thought that although justified for the sake of self consistency, detailing the EDCs did not bring any insight into the question addressed here and increased significantly unnecessary mathematical notation. Nonetheless we acknowledge the comment made by referee 1 and we have made substantial changes to section

2.2 to incorporate a complete description of the EDCs, running from line 123 to line 180 in the revised manuscript.

**1.4 Other comments**

A discussion section has been added to the manuscript to address the remaining comments.

Global scale experiments are discussed in reference to Rayner et al. (2005) and CCDAS. The work of Kemp et al. (2014), which directly relates to what we discuss in our paper, is also cited. Although this preliminary work is not reported in our manuscript the three approaches presented in Kemp et al. (2014) were evaluated for the preparation of the manuscript, and our conclusion was different: in our case incorporating the EDCs by adding a penalty term to the cost function was the most successful approach to constrain unresolved parameters and, most importantly in our case, to allow for a better uncertainty quantification.

We also refer to Ziehn et al. (2012) for their comparison between MCMC method and 4DVAR in CCDAS. A MCMC method is used in Bloom and Williams (2015), and in Safta et al. (2015), added in reference to the revised manuscript, a detailed analysis of MCMC for DALEC is performed. A comparison between MCMC and 4DVAR was beyond the scope of this paper, our intention was rather to establish 4DVAR as a suitable method for DALEC and the EDCs. Nonetheless a comparison between 4DVAR and fully non-linear methods is necessary, it is one of the aspects of our current work.

Finally we mention our current work on a hybrid ensemble-variational method. This approach provides an adjoint-free formulation of the variational problem and show promising results in the context discussed in the manuscript. This work is part of a paper in preparation.

**2  Response to referee 2**

The version of DALEC used for this study, referred to as DALECv2 and described in details in Bloom and Williams (2015), is a nonlinear dynamical system, it is not a linear autonomous system. The trajectories of the carbon pools $\boldsymbol{C}$ are computed using the recursion formula

$$\boldsymbol{C}^{t+1} = \boldsymbol{C}^t + \boldsymbol{f}(\boldsymbol{C}^t, \boldsymbol{p}, \phi(t))\Delta t, \tag{1}$$

where $\boldsymbol{f}$ is a nonlinear vector valued function of the carbon pools, the parameters $\boldsymbol{p}$ and the meteorological drivers $\phi(t)$, $\Delta t$ denoting the step time in month in our case. The nonlinear nature of the model is stressed out in the revised manuscript at lines 77 and 83. Moreover, the definition of $\boldsymbol{h}$ is clarified at line 197.

The main focus of the paper is on the vector $\boldsymbol{x} = \log([\boldsymbol{p}, \boldsymbol{C}_0])^T$. In section 2.3 first where we investigate the sensitivity of different outputs with respect to $\boldsymbol{x}$ and its components, and then in subsequent sections where $\boldsymbol{x}$ is estimated using inverse methods. The vector $\boldsymbol{x}$, denoting fixed quantities as initial conditions and parameters for the dynamical system DALECv2, is seen as the variable from the

point of view of sensitivity analysis and inverse modelling and therefore its components are referred to as state variables, input variables or parameters interchangeably throughout the manuscript. This choice of terminology, stressed out at line 228, have been reinforced in the revised manuscript by adding the present paragraph at line 103.

[revised manuscript text omitted]

LAI, NEE, GPP and RESP. LAI monthly mean observations for Ameriflux sites are extracted from MOD15A2 LAI 8-day version 005 1km-resolution product. These observations together with the meteorological drivers are provided by A. Bloom and J. Exbrayat: details about their construction can be found in Bloom and Williams (2015). At Ameriflux sites we use the level 4 data product

100 (available at http://cdiac.ornl.gov/ftp/ameriflux/data/Level4/), which provides monthly means for NEE and GPP. NEE and GPP are then used to define total respiration (RESP) as RESP=NEE+GPP. The meteorological drivers span over a period of twelve years from 2001 to 2013. LAI observations are available during the full period but for NEE and GPP, and thus RESP, shorter  records are available depending on the Ameriflux site. In this study we consider the Morgan Monroe state

105 forest located in Indiana, US (39.3,-86.4). This Ameriflux site is composed in majority of mixed hardwood broadleaf deciduous trees and classifies as a humid subtropical climate.

In the remainder of the paper the main focus is on the vector $x = \log([p, C_0])^T$: in section 2.3 first where we investigate the sensitivity of different outputs with respect to $x$ and its components, and then in subsequent sections where $x$ is estimated using inverse methods. The vector $x$, denoting

110 fixed quantities as initial conditions and parameters for the dynamical system DALECv2, is seen as the variable from the point of view of sensitivity analysis and inverse modelling and therefore its components will be referred to as state variables, input variables or parameters interchangeably throughout the manuscript.

**2.2 Ecological constraints**

115 Over the last decade many inverse  modelling studies have used NEE measurements from the fluxnet network, together with other types of observations when available, to provide information about processes controlled by parameters with respect to which NEE is weakly sensitive. Though it contains an  ever-increasing amount of information, the flux tower network only provides sparse coverage of terrestrial ecosystems. On the other hand, despite a good spatial and tem-

120 poral coverage, MODIS LAI monthly mean observations only constrain a limited set of DALECv2 state variables, and additional information is required in order to  regularise the ill-posed problem and obtain a meaningful solution.

Additional information can be obtained by imposing priors on the variables or by adding other observation streams (biomass, soil organic matter, ...). As an alternative, Bloom and Williams introduced

125 a set of constraints, referred to as ecological and dynamical constraints (EDCs). These constraints, detailed in Bloom and Williams (2015) can be divided into two groups: static and  dynamic constraints. The static constraints which directly impose conditions on the parameters are:

- turnover rates constraints which ensure that turnover rates ratios are consistent with knowledge of the carbon pools residence times.

$$\text{EDC}_1: \quad p_9 < p_8, \tag{1}$$

$$\text{EDC}_2: \quad p_9 < p_1, \tag{2}$$

$$\text{EDC}_3: \quad p_6 < 1/(p_5 \times 365.25), \tag{3}$$

$$\text{EDC}_4: \quad p_7 > p_9 \exp p_{10} \bar{T}, \tag{4}$$

$$\text{EDC}_5: \quad p_{12} + 45 < p_{15}, \tag{5}$$

where $\bar{T}$ denotes the mean temperature within the drivers time window. $\text{EDC}_4$ is a modification to the constraint proposed in Bloom and Williams (2015), it is currently used in the CARDAMON framework (http://www.geos.ed.ac.uk/homes/mwilliam/CARDAMOM.html).

- Root-foliar allocation which allows for a strong correlation between parameters controlling allocation to foliage and roots.

$$\text{EDC}_6: \quad f_{\text{root}} < 5(f_{\text{fol}} + f_{\text{lab}}), \tag{6}$$

$$\text{EDC}_7: \quad f_{\text{fol}} + f_{\text{lab}} < 5 f_{\text{root}}, \tag{7}$$

where the allocation fractions $f_{\text{fol}}$, $f_{\text{lab}}$ and $f_{\text{root}}$ are defined by

$$f_{\text{auto}} = p_2, \tag{8}$$

$$f_{\text{fol}} = (1 - f_{\text{auto}})p_3, \tag{9}$$

$$f_{\text{lab}} = (1 - f_{\text{auto}} - f_{\text{fol}})p_{13}, \tag{10}$$

$$f_{\text{root}} = (1 - f_{\text{auto}} - f_{\text{fol}} - f_{\text{lab}})p_4. \tag{11}$$

The dynamic constraints, for which a model run is performed to define attractors, limit the application of the model to ecosystems with no major recent disturbance. They are defined by:

- Root-foliar mean dynamics

$$\text{EDC}_8: \bar{C}_{\text{r}} < 5\bar{C}_{\text{f}}, \tag{12}$$

$$\text{EDC}_9: \bar{C}_{\text{f}} < 5\bar{C}_{\text{r}}, \tag{13}$$

where $\bar{C}_{\text{f}}$ and $\bar{C}_{\text{r}}$ denote the mean of $C_{\text{f}}$ and $C_{\text{
[revised manuscript text omitted]